# TTYH family members form tetrameric complexes at the cell membrane

Emelia Melvin [1], Zuzana Kalaninová[2,3], Elia Shlush[1], Petr Man [2], Moshe Giladi [1,4 ✉] & Yoni Haitin[1,5 ✉]

The conserved Tweety homolog (TTYH) family consists of three paralogs in vertebrates, displaying a ubiquitous expression pattern. Although considered as ion channels for almost two decades, recent structural and functional analyses refuted this role. Intriguingly, while all paralogs shared a dimeric stoichiometry following detergent solubilization, their structures revealed divergence in their relative subunit orientation. Here, we determined the stoichiometry of intact mouse TTYH (mTTYH) complexes in cells. Using cross-linking and single-molecule fluorescence microscopy, we demonstrate that mTTYH1 and mTTYH3 form tetramers at the plasma membrane, stabilized by interactions between their extracellular domains. Using blue-native PAGE, fluorescence-detection size-exclusion chromatography, and hydrogen/deuterium exchange mass spectrometry (HDX-MS), we reveal that detergent solubilization results in tetramers destabilization, leading to their dissolution into dimers. Moreover, HDX-MS demonstrates that the extracellular domains are stabilized in the context of the tetrameric mTTYH complex. Together, our results expose the innate tetrameric organization of TTYH complexes at the cell membrane. Future structural analyses of these assemblies in native membranes are required to illuminate their long-sought cellular function.

[1] Department of Physiology and Pharmacology, Sackler Faculty of Medicine, Tel Aviv University, Tel-Aviv 6997801, Israel. [2] Institute of Microbiology of the Czech Academy of Sciences, Division BioCeV, Prumyslova 595, 252 50 Vestec, Czech Republic. [3] Department of Biochemistry, Faculty of Science, Charles University, Hlavova 2030/8, 128 43, Prague 2, Czech Republic. [4] Tel Aviv Sourasky Medical Center, Tel Aviv 6423906, Israel. [5] Sagol School of Neuroscience, Tel Aviv University, Tel Aviv 6997801, Israel. ✉email: moshegil@tauex.tau.ac.il; yhaitin@tauex.tau.ac.il

The Tweety protein (TTY) was originally identified in the *Drosophila melanogaster* flightless locus[1,2]. Subsequent studies identified three conserved TTY homologs (TTYH1-3) in vertebrates[3–5], exhibiting a differential expression pattern. While TTYH3 is ubiquitously expressed[6], the expression of TTYH1 and TTYH2 is limited to the central nervous system[7] and testis[8]. Initial studies using *Drosophila* knockout did not establish a clear functional role for this family[3]. However, later reports suggested that these conserved membrane proteins serve as $Cl^-$ channels, activated by either $Ca^{2+}$ or cell swelling[6,9]. Specifically, it was suggested that TTYH1 mediates $Cl^-$ currents in response to mechanical force[6,10], while TTYH2 and TTYH3 paralogs underlie a $Ca^{2+}$-dependent maxi-$Cl^-$ current found in neurons[11] and skeletal muscle[12]. Moreover, TTYH-mediated $Cl^-$ currents were recorded using mammalian heterologous expression systems[6,13].

Importantly, two recent studies elucidated the long-sought structure of TTYH family members[14,15]. However, strikingly, anion conductive pathways were not identified in any of the human (hTTYH) or mouse (mTTYH) members. Moreover, functional studies failed to reproduce the previously reported ion channel activity related to TTYH expression[9]. The structures revealed a novel subunit topology common to all TTYH members, consisting of five transmembrane helices and a large α-helical extracellular domain (ECD). hTTYH paralogs and mTTYH3 were shown to exhibit a side-by-side *cis*-dimeric organization, involving interaction interfaces within the ECD and transmembrane regions[14,15]. This is in contrast to the predicted tetrameric organization inferred from electrophysiological studies[9]. Interestingly, in the absence of $Ca^{2+}$ in the solution, mTTYH2 demonstrated a different organization, consisting of a head-to-head *trans*-dimeric interaction between the ECDs of subunits from juxtaposing membranes[15].

Unfortunately, despite the extensive and detailed structural characterization of five different TTYH family members, their functional role remains debated[16]. Nonetheless, previous studies demonstrated that members of this family are involved in several pathophysiological processes[17]. For example, TTYH members were shown to dramatically change their expression pattern following Status Epilepticus[18], as well as to drive the colonization of gliomas[19]. Indeed, *ttyh1* was shown to be strongly upregulated in an array of childhood brain tumors[7], with fusion of its promoter to the large microRNA cluster C19MC driving the development of a particularly aggressive form of pediatric cancer[20].

As the structures of TTYH members, determined following detergent-mediated membrane solubilization, revealed distinct spatial organizations[14,15], elucidation of their stoichiometry in the native membrane environment may hint toward their functional and cellular roles. Indeed, the extraction of membrane proteins from their native environment can result in dramatic changes to their ternary and quaternary organizations[21], hampering the

elucidation of TTYH physiological roles. Therefore, we sought to determine the stoichiometry of TTYH under native conditions. Unexpectedly, using biochemical analyses, together with single-molecule fluorescence microscopy[22], fluorescence-detection size-exclusion chromatography (FSEC)[23], and size-exclusion chromatography multi-angle light scattering (SEC-MALS), we show that TTYH family members form tetrameric complexes in the native membrane environment, which dissociate into dimers by detergent solubilization.

## Results

### mTTYH members form high-order oligomeric assemblies in cells.

To estimate the stoichiometry of TTYH proteins within their native membrane environment, we overexpressed FLAG-tagged mTTYH paralogs in HEK 293 cells and performed western blot analysis (Fig. 1a). While we could clearly identify the expression of mTTYH1 and mTTYH3, a band corresponding to mTTYH2 could not be detected. Hence, we proceeded with the investigation of mTTYH1 and mTTYH3 complex stoichiometry using in situ cross-linking with the amine-reactive, homo-bifunctional bis-sulfosuccinimidyl suberate ($BS^3$)[24,25]. Importantly, $BS^3$ is membrane impermeable, and thus can only cross-link primary amines of mTTYH subunits at the cell surface, exposed to the extracellular milieu (Supplementary Fig. 1). Incubation of the cells with $BS^3$ resulted in the emergence of multiple migration species in both paralogs, with a mass culminating to that expected for tetrameric oligomers (~240 kDa) (Fig. 1b, c). Together, these results suggest that mTTYH proteins may form higher-order oligomeric assemblies under native conditions and point toward a possible common tetrameric assembly for mTTYH1 and mTTYH3.

### Tetrameric assembly of mTTYH complexes revealed by single-molecule subunit counting.

While in situ cross-linking indicated that mTTYH proteins are composed of multiple subunits, we cannot exclude the possibility that heteromeric assembly formation of mTTYH dimers[14,15] with additional endogenous proteins underlies the observed migration patterns. Thus, to directly examine the number of mTTYH subunits within a complex in living cells, we resorted to the single-molecule subunit counting approach[22]. This method directly inspects the number of subunits within membrane-delimited complexes by monitoring the quantal bleaching of conjugated fluorophores. To this end, we generated C-terminal EGFP fusions of mTTYH1 and mTTYH3. Next, we titrated mTTYH-EGFP expression in *Xenopus* oocytes until a low enough level was achieved, enabling us to resolve individual spots using Total Internal Reflection Fluorescence Microscopy (TIRFM) (Fig. 2a, d and Supplementary Movie 1).

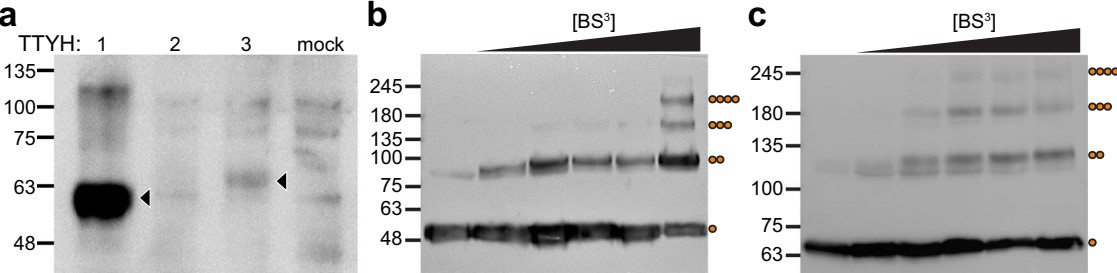

**Fig. 1 In situ cross-linking analyses of mTTYH family members. a** SDS-PAGE western blot analysis of HEK 293 cells transiently transfected with the indicated FLAG-tagged mTTYH paralogs. Arrowheads indicate the bands corresponding to mTTYH1 and mTTYH3. Note that there is a nonspecific band in mTTYH2, which can also be found in non-transfected cells (mock). **b, c** SDS-PAGE western blot of mTTYH1 (**b**) and mTTYH3 (**c**) cross-linking with increasing concentrations of $BS^3$ (concentrations used were 0, 10, 50, 100, 200, and 2000 μM from left to right). Molecular weights and oligomeric states are shown on the left and right, respectively.

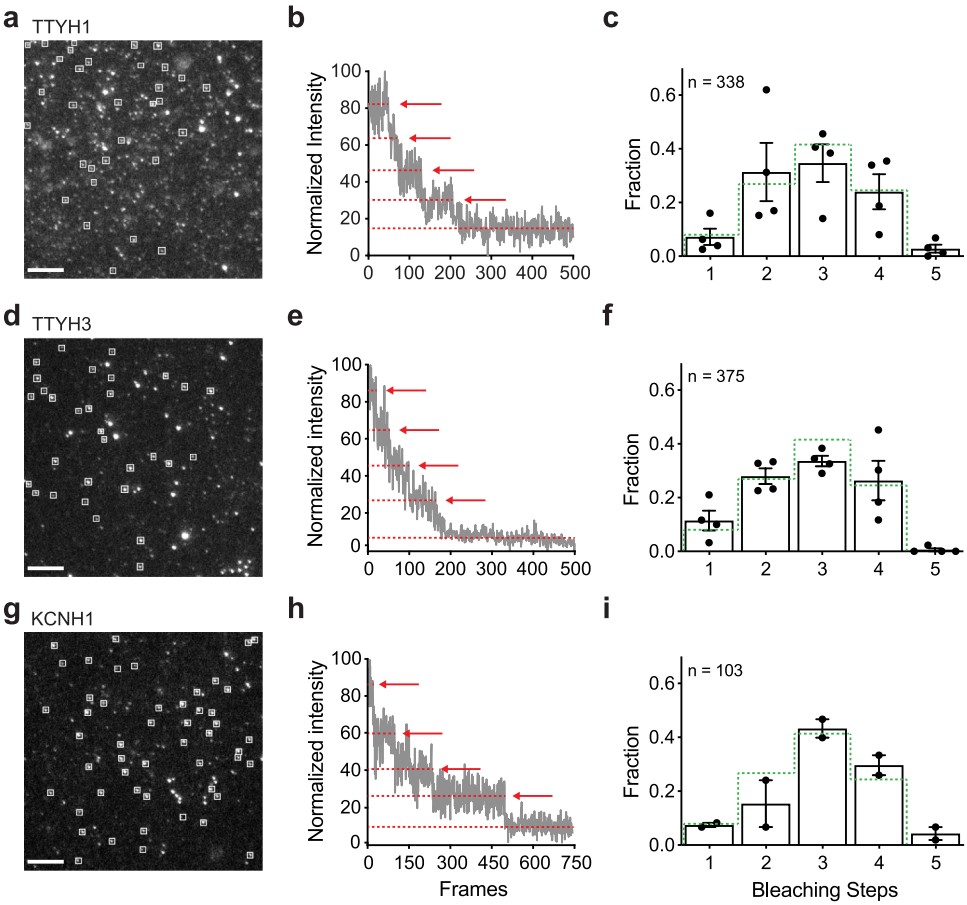

**Fig. 2 Single-molecule subunit counting of mTTYH family members.** Analyses of EGFP-fused mTTYH1 (**a–c**), mTTYH3 (**d–f**), and KCNH1 (**g–i**). **a**, **d**, **g** Representative images sampled from 2 min movies, 1-day post-injection of 1 ng cRNA, using TIRFM. Spots of interest are highlighted by white squares. Scale bar, 2 µm. **b**, **e**, **h** Representative fluorescence intensity traces obtained from single spots showing four steps of photobleaching, as indicated by arrows. **c**, **f**, **i** Probability distributions of bleaching steps (black bars), with error bars indicating SEM. Binomial distribution ($p = 0.7$) for tetrameric complexes is presented as a dashed green line. n indicates the number of spots analyzed.

Cells were continuously illuminated, and time-series images were recorded at 30 Hz until reaching baseline (10–20 sec). Fluorescence intensities of stationary fluorescent spots were measured over the illumination duration, and bleaching steps were counted manually (Fig. 2b, e). As a positive control, we used the EGFP fusion of KCNH1, a known tetrameric voltage-gated $K^+$ channel[26] (Fig. 2g–i). Importantly, both mTTYH1 and mTTYH3 exhibited bleaching profiles similar to that of KCNH1 (Fig. 2c, f, i), consistent with their tetrameric assembly. Indeed, the data were best fit by a binomial distribution[27] assuming a 4-mer with a probability ($p$) of EGFP being fluorescent of 0.7 (Supplementary Fig. 2). Importantly, the stoichiometry we measured by subunit counting agrees well with the in situ cross-linking results.

**Detergent solubilization of mTTYH complexes results in the emergence of a dimeric population.** In contrast to the tetrameric stoichiometry we observed in the cellular environment, the recent structures of members of the TTYH family shared a dimeric organization[14,15]. However, as these structures were determined following membrane solubilization by detergents, we hypothesized that this process may result in the dissociation of TTYH complexes. Therefore, we used Blue-Native PAGE (BN-PAGE) to assess subunit stoichiometry following detergent solubilization using dodecyl maltoside (DDM), most often used for the solubilization and purification of membrane proteins (Fig. 3a). This analysis revealed that both mTTYH1 and mTTYH3 display a complex migration pattern,

consisting of dimeric (~120 kDa) and tetrameric (~240 kDa) populations (Fig. 3a), indicating detergent-mediated perturbation of the quaternary structure of the complex. Conversely, the dimeric population may arise from the reagents used for the BN-PAGE analysis (e.g. Coomassie stain)[28]. Thus, to examine whether the dimeric population stems from a detergent-mediated tetramers disassembly, we used FSEC. HEK 293 cells expressing EGFP fusions of mTTYH1 or mTTYH3 were solubilized with DDM and subjected to FSEC analyses (Fig. 3b). Intriguingly, poly-dispersed elution profiles were evident for both paralogs, with peaks detected at volumes corresponding to tetrameric as well as dimeric assemblies. Specifically, mTTYH1 displayed an elution profile mainly corresponding to that expected for dimers, while mTTYH3 dispersion was skewed toward an elution volume expected for tetramers (Fig. 3b).

Next, to validate the oligomeric state of TTYH in each FSEC elution peak, we proceeded with purification of intact mTTYH1-GFP for SEC-MALS analysis (Fig. 3c). Indeed, the peaks eluting at the volumes assumed to represent tetramers and dimers revealed a protein mass of $223 \pm 0.4$ and $109 \pm 1.4$ kDa for the tetrameric and dimeric populations, respectively. As expected, the mass of the tetrameric population is double that of the dimeric one, and both are within the experimental error of the MALS measurement.

To gain further insights into the dissociation of mTTYH tetramers to dimers, we studied their thermal denaturation using a FSEC-based thermostability assay (FSEC-TS)[29]. Samples were subjected to 10 min thermal insults, ranging from 20 °C to 80 °C,

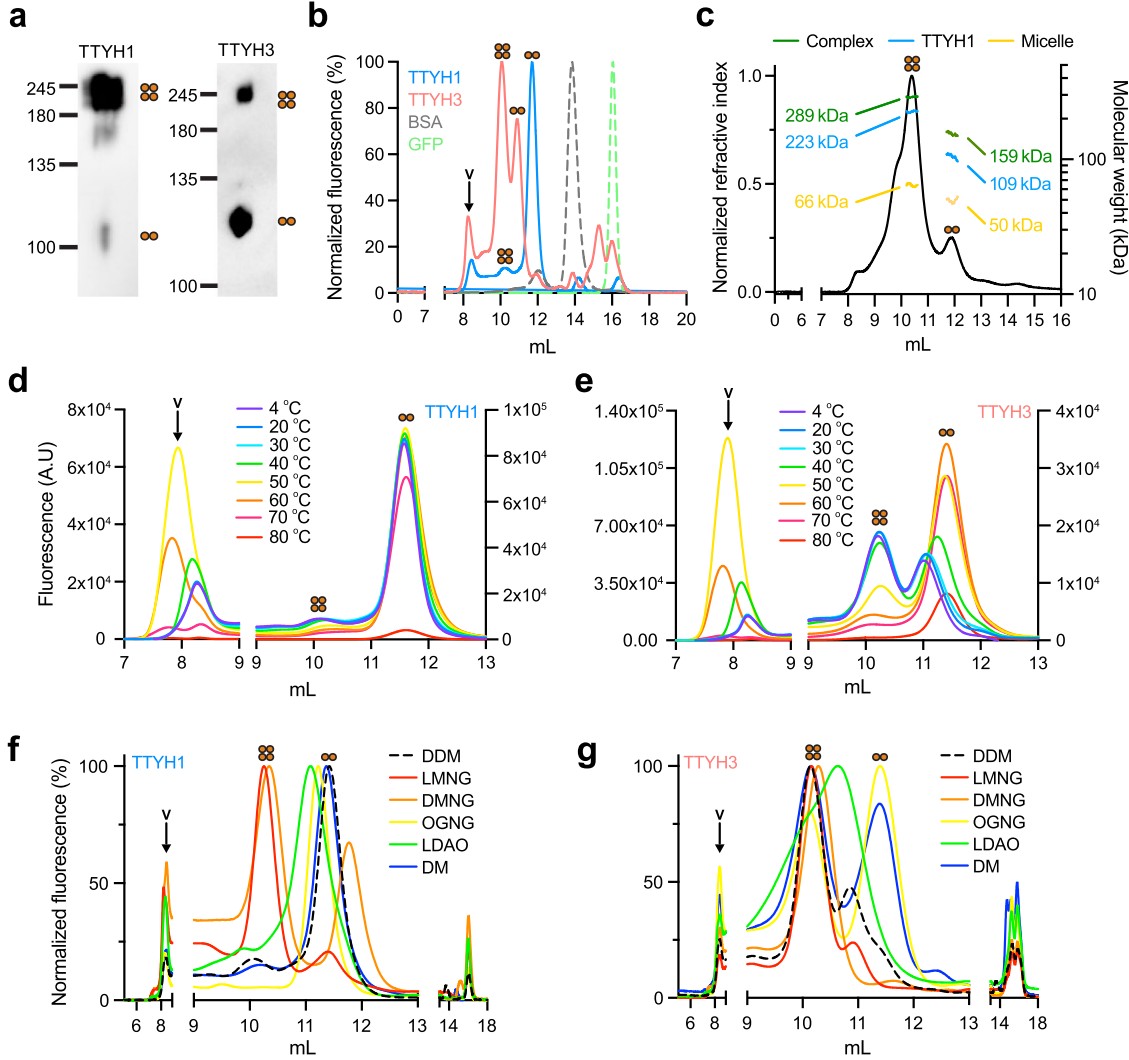

**Fig. 3 Analyses of detergent-solubilized mTTYH paralogs. a** BN-PAGE western blot analysis of HEK 293 cells transiently transfected with mTTYH1 (left) or mTTYH3 (right). **b** FSEC chromatograms depicting the elution profiles of DDM solubilized HEK 293 cells expressing the indicated EGFP-fused mTTYH paralogs. BSA and EGFP profiles are provided for reference. **c** Purified mTTYH1-GFP SEC-MALS analysis. The normalized refractive index is shown as a black line, while the calculated masses of the protein, detergent, and protein–detergent complex are shown as blue, yellow, and green lines, respectively. **d**, **e** FSEC chromatograms of mTTYH1 (**d**) and mTTYH3 (**e**) preincubated for 10 min at the indicated temperatures. **f**, **g** FSEC chromatograms of mTTYH1 (**f**) and mTTYH3 (**g**), solubilized using different detergents, as indicated. Orange circles represent the oligomeric states of the complex. Column void volume (v) is indicated by arrow where applicable.

followed by centrifugation and FSEC analysis (Fig. 3d, e). With increasing temperatures, the population distribution shifted toward dimers (Fig. 3d, e and Supplementary Fig. 3), accompanied by the emergence of soluble aggregates eluting at the column void volume. These results suggest that mTTYH complexes may feature a dimer-of-dimers assembly, with a detergent-resistant dimeric interface and a labile surface involved in tetramerization.

The properties of the detergent used for membrane protein solubilization were previously shown to affect the oligomeric stoichiometry in different proteins[21]. Thus, we sought to determine whether solubilization with different detergents may result in an altered mTTYH tetramer to dimer ratio (Fig. 3f, g). We focused on a detergent screen designed to test the effect of the polar headgroup properties and the alkyl chain lengths on maintaining mTTYH tetrameric complex integrity. Specifically, following solubilization of the cells with lauryl dimethylamine-N-oxide (LDAO), a detergent sharing an identical alkyl chain length with DDM, but containing a charged amine oxide headgroup, mTTYH1 displayed an elution profile reminiscent of that

obtained with DDM, while mTTYH3 was partially skewed toward dimers. Thus, the use of a charged headgroup did not significantly affect the tetramer to dimer ratio. Conversely, the use of detergents of the neopentyl glycol (NG) class, containing two alkyl chains of varying lengths, revealed a positive correlation between the length of the alkyl chains and the emergence of a tetrameric population. Indeed, membrane solubilization with lauryl maltose neopentyl glycol (LMNG; $C_{12}$ chains) resulted in a near-complete tetrameric shift in both paralogs, while the use of octyl glucose neopentyl glycol (OGNG; $C_8$ chains) failed to produce a similar effect, and decyl maltose neopentyl glycol (DMNG; $C_{10}$ chains) had an intermediate effect (Fig. 3f, g). Supporting the contribution of the alkyl chain length to tetramer stabilization, solubilizing the cells using decyl maltoside (DM), a detergent with a short alkyl chain, skewed the population distribution toward dimers. Together, these results demonstrate the negative impact of detergents with short alkyl chains on the solubilization of the native TTYH complex and highlight the detergent-mediated destabilization of their tetrameric assembly.

**HDX-MS analyses of mTTYH complexes**. In order to identify protein regions destabilized by detergent-mediated tetramer dissociation, we compared the HDX-MS profiles of the dimeric and tetrameric populations of mTTYH1, solubilized with DDM. In this method, following proteolytic fragmentation, the rate of backbone HDX of each peptide is determined, providing information on the secondary structure and solvent accessibility of protein sub-regions[30]. Accordingly, it is expected that the protein core, secondary structure elements, and buried interfacial regions will exhibit low HDX. The HDX profiles were obtained at 25 °C, following exchange for 20 to 7200 s (Supplementary Fig. 4). However, most peptides exhibited very high deuteration levels throughout the experiment, hindering a meaningful comparison of the backbone dynamics between the two populations. Therefore, we repeated the experiments at 4 °C, following exchange periods of 2 to 180 s (Fig. 4). Under these conditions, the HDX kinetics of the identified peptides were well resolved (Fig. 4c). Previous studies indicated that TTYH dimerization is mediated by an extensive interfacial region, which includes TM2 and TM5 transmembrane helices, as well as E5 and E6 ECD helices (Fig. 4a, c)[14,15]. Surprisingly, mapping the HDX profile of the dimeric population onto a homology model of mTTYH1[31] revealed relatively high deuterium incorporation at the proposed dimeric interface compared with non-interfacial regions (Fig. 4a, c). Specifically, while the TM2 helix, positioned at the core of the dimeric complex, showed high HDX levels, the peripheral TM1 helix showed exceptionally low exchange throughout the time course tested. At the ECD, the peripheral helices E1, E2, and E3 exhibited lower deuterium uptake than the interface-forming helices E5 and E6 (Fig. 4a, c). Of note, as anticipated, low uptake was observed in the well-folded fused GFP moiety (Supplementary Fig. 5a, c). Thus, the unexpected HDX profile of mTTYH1 suggests that the dimeric interface observed in the cryo-EM structures is highly exposed to the bulk solvent and/or misfolded, reflecting the detergent-mediated destabilization of the native assembly. Intriguingly, analysis of the tetrameric population revealed an overall similar HDX profile (Fig. 4a, c), suggesting a similar detergent-mediated destabilization of this population. Nevertheless, analysis of the ΔHDX between the tetrameric and dimeric populations highlighted lower deuterium uptake at the N- and C-termini, and the ECD of the tetrameric form (Fig. 4b, c), with the largest differences observed at the E1-E2 connecting loop, as well as E4 and E5 helices. Notably, the tetrameric population exhibited an overall reduced deuterium uptake also at 25 °C. No differences were observed in the fused GFP moiety under both experimental settings (Supplementary Fig. 5b, c). Thus, the ΔHDX indicates the possible role of the ECD in TTYH tetramerization.

**The tetrameric interface of mTTYH is stabilized by intra-subunit disulfides**. Our HDX-MS results implicate the ECD in tetramerization. The recent structures of both hTTYH and mTTYH paralogs revealed large interaction interfaces between the ECDs[14,15]. Moreover, the structures show that the ECD contains two disulfide bridges stabilizing a helical flap-like hairpin fold formed between conserved cysteine residues (Fig. 5a). One of these bridges (C303-C370 in mTTYH1, corresponding to C299-C366 in mTTYH3), demonstrates reduced HDX in the tetramer, highlighting its possible contribution to the tetrameric assembly. We reasoned that if the ECD is important for tetramerization, perturbation of these conserved disulfide bridges may result in destabilization of the tetrameric interaction interfaces between the subunits. Therefore, following solubilization with DDM, FSEC was performed before and after incubation with 20 mM 2-mercaptoethanol (2-ME) (Fig. 5b). Importantly, 2-ME resulted in a shift toward dimers (Fig. 5c). As the tetrameric population of mTTYH1 is relatively small, this shift was more apparent in mTTYH3. Nevertheless, a similar effect was observed following

solubilization with LMNG, which enriches the tetrameric population (Fig. 3f, g and Supplementary Fig. 6). To further pinpoint the disulfide bridge contributing to tetramer stabilization, we proceeded with analyses of mTTYH3-GFP ECD cysteine mutants' impact on tetramerization (Fig. 5d). Akin to the effect of 2-ME, the FSEC elution profile of C299A and C366A (Fig. 5e) resulted in a marked shift toward dimers (Fig. 5f). These data further support the notion that the dimers arise from the dissociation of tetramers, highlighting the role of the ECDs in complex assembly.

**Isolated mTTYH3 ECDs form tetramers in solution**. The HDX-MS analysis of intact mTTYH1 (Fig. 4), along with the effect of 2-ME (Fig. 5), point toward the contribution of the ECD to tetramerization. Thus, to establish the role of the ECDs in tetramerization, we designed a construct consisting of the isolated domains (Fig. 6a). Specifically, focusing on mTTYH3, we introduced a T4 lysozyme (T4L) sequence between ECD1 (positions 107-209) and ECD2 (positions 260-386) to increase protein stability and solubility[32]. Due to the presence of disulfide bridges and glycosyl moieties[14,15,33], we expressed this FLAG-tagged construct as a secreted protein in insect cells to allow intracellular trafficking for proper post-translational modifications processing. Indeed, western blot analysis of the cell lysate revealed a complex migration pattern consistent with different stages in the protein biogenesis pathway. Conversely, a single band was detected in the secreted fraction obtained from the clarified medium (Fig. 6b, left panel). Indeed, the secreted fraction is glycosylated, as determined by the change in the band mobility following treatment with Peptide:N-glycosidase (PNGase) F (Fig. 6b, right panel). The secreted protein was subjected to cross-linking analysis using the homobifunctional amine-reactive cross-linker disuccinimidyl suberate (DSS), resulting in the emergence of higher-order oligomers, culminating in the size expected for a tetramer (Fig. 6c). Thus, together with the results obtained using intact TTYH (Fig. 5), we suggest that the ECDs participate in TTYH tetramerization.

## Discussion
Over the past years, the TTYH family gained attention mainly due to its expression in the CNS[7,8,34] and its association with various pathological conditions[18–20,34]. These observations motivated functional and structural studies[17], culminating in the recent structural elucidation of hTTYH and mTTYH members[14,15]. Intriguingly, these structures lack an obvious Cl⁻ conductive pathway, contradicting the originally assigned functional role of these proteins as ion channels and rendering the functional role of the TTYH family enigmatic. Here, we performed a comprehensive analysis of mTTYH stoichiometry in their natural settings.

Our data suggest that mTTYHs exist as tetramers at the cell membrane and expose the impact of detergent solubilization on complex integrity. Indeed, in situ cross-linking with a membrane-impermeable cross-linker (Supplementary Fig. 1) resulted in an SDS-PAGE migration pattern consistent with the formation of tetramers at the cell surface (Fig. 1b, c). Moreover, to exclude the possibility that the molecular weight increase reflects the association of mTTYH subunits with endogenous proteins[35–39], we directly assessed the number of subunits within mature complexes in their undisrupted environment using single-molecule fluorescence microscopy approaches (Fig. 2). Consistent with the tetrameric stoichiometry detected using cross-linking measurements, photobleaching steps distribution in subunit counting experiments[22], for both paralogs, was in striking agreement with that measured for KCNH1 (Fig. 2), a K⁺ channel with well-established tetrameric stoichiometry[26]. Together, our investigations unequivocally argue that four mTTYH subunits assemble to form

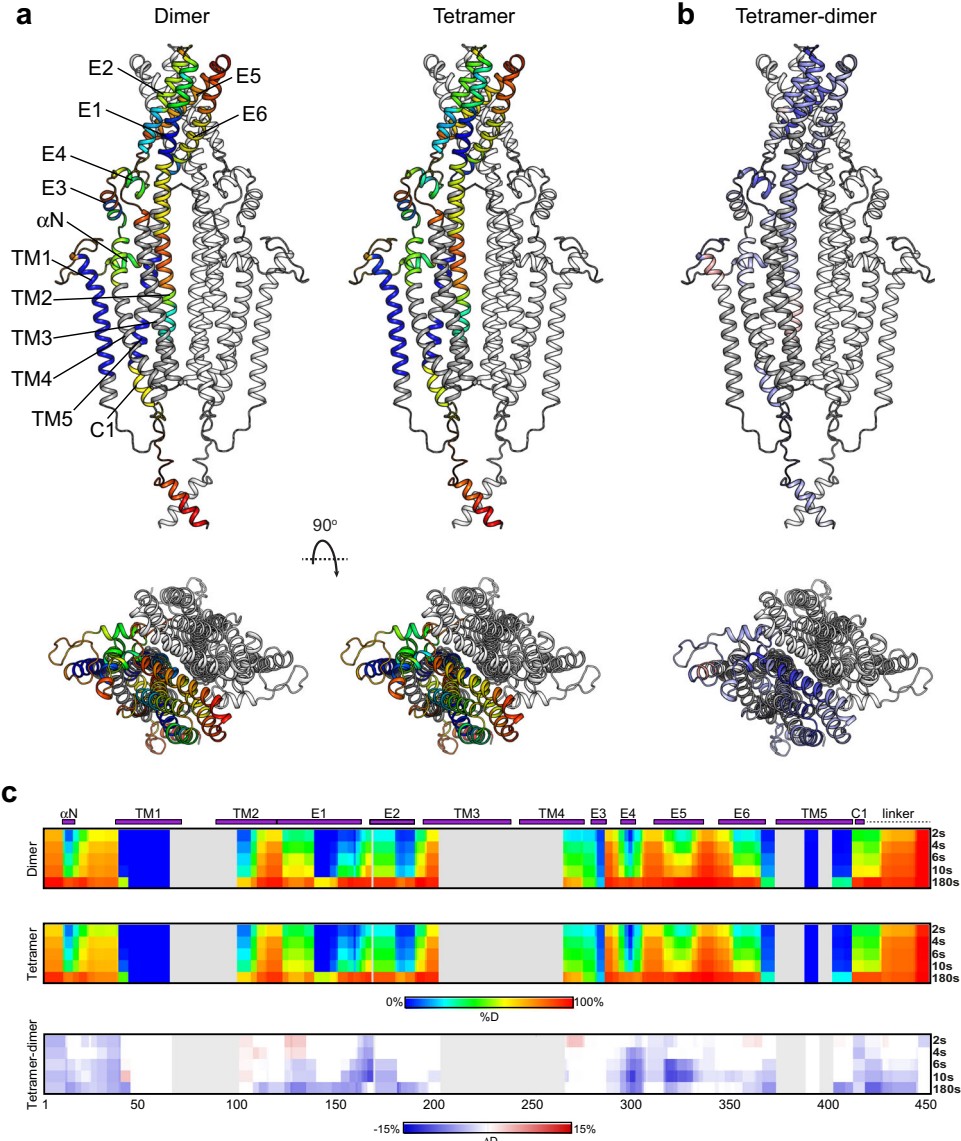

**Fig. 4 HDX-MS analysis of mTTYH1. a** Cartoon representations of mTTYH1 homology model. A single chain is colored as a heat map, representing the HDX level following 10 sec (s) incubation in $D_2O$ for the dimeric (left) and tetrameric (right) populations. The highest and lowest exchange levels are indicated in red and blue, respectively. **b** The HDX difference between the tetrameric and dimeric populations, following 10 s incubation in $D_2O$, is mapped onto the mTTYH1 homology model. **c** Deuteration levels at the indicated time points for the dimer (upper panel) and tetramer (middle panel). The difference between the tetrameric and dimeric populations is shown (bottom panel). The secondary structure elements are indicated above the heat maps.

the native complexes and that the high molecular weight bands observed using the in situ cross-linking analysis indeed represent mTTYH homotetramers.

The recent cryo-EM structures of TTYH family members revealed a common dimeric stoichiometry, in sharp contrast to the tetrameric assemblies observed here in the cellular milieu. Puzzled by this apparent discrepancy, we sought to determine its potential causes. The key difference between the experimental systems is the use of detergents in the process needed for obtaining a well-behaving and purified sample for cryo-EM studies, in contrast to the native environment of the cellular membrane. Indeed, detergent solubilization was previously shown to dramatically alter the quaternary organization, stoichiometry, and function of various membrane proteins. For example, previous BN-PAGE and FSEC analyses of the bona fide tetrameric GluA1 and GluA2 revealed that tetrameric and dimeric populations coexist[40], attributed to their dimer of dimers assembly mode[41].

Similarly, recent cryo-EM studies of the membrane receptor Patched 1 (Ptch1) suggested monomeric[42], dimeric[43], or tetrameric[44] stoichiometries, when determined in DDM micelles, amphipols, or GDN, respectively. This is a stark example of the potential discrepancies in the oligomeric states of membrane proteins subjected to different solubilizing agents. Indeed, BN-PAGE and FSEC analyses of DDM-solubilized mTTYH complexes expose that in addition to the tetrameric population, a significant dimeric fraction was evident (Fig. 3a, b). Importantly, SEC-MALS analysis of purified mTTYH1-GFP verified the presence of tetrameric and dimeric populations following detergent solubilization (Fig. 3c). Moreover, using FSEC-TS, we observed a temperature-related shift in the stoichiometry of both mTTYH1 and mTTYH3, solubilized in DDM (Fig. 3d, e). Finally, solubilization using detergents varying in their polar headgroup and alkyl chains showed that the tetrameric complexes are preferentially stabilized by detergents with increasingly long alkyl

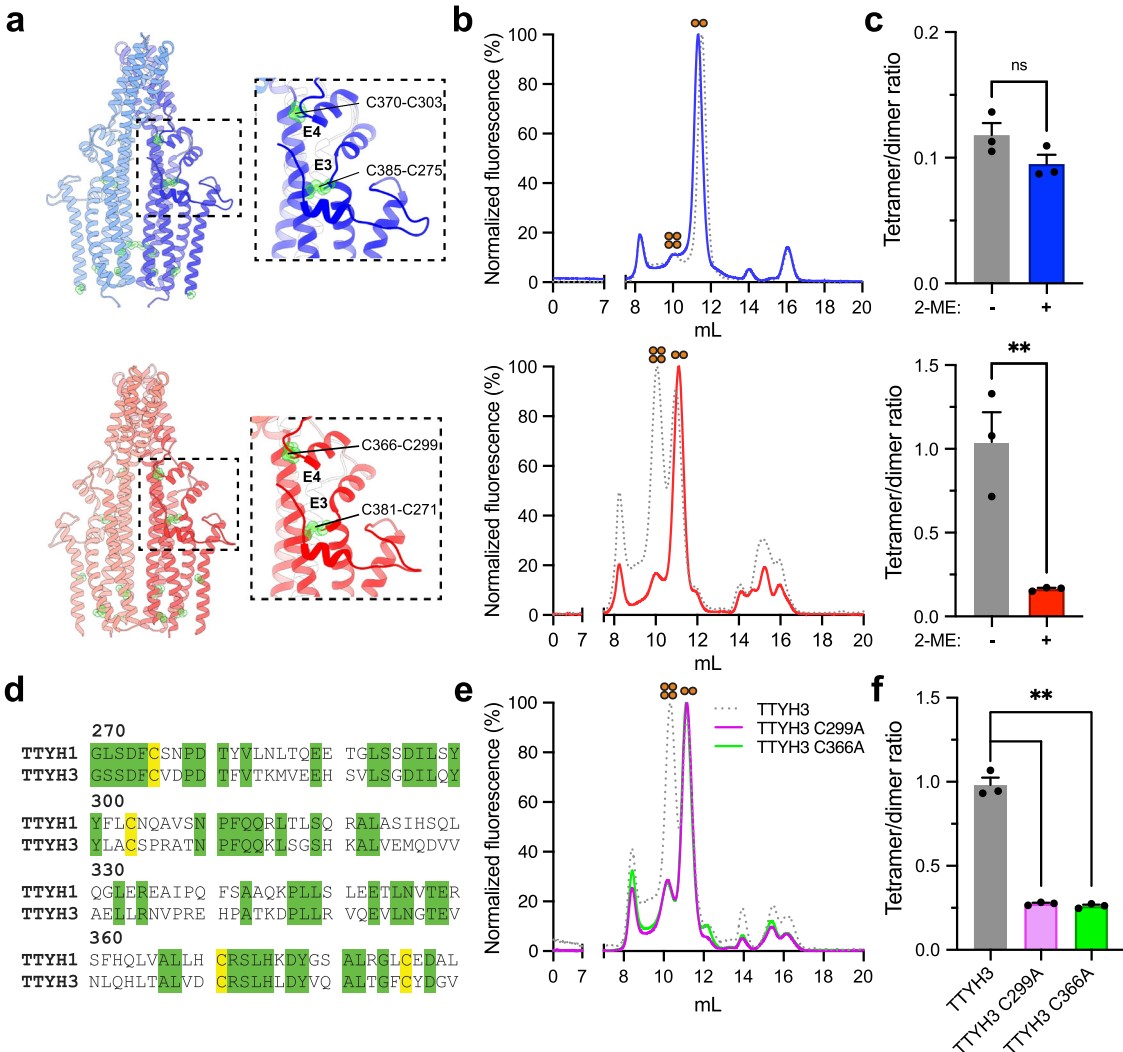

**Fig. 5 mTTYH disulfides contribute to tetramerization. a** Cryo-EM structures of hTTYH1 (upper panel; PDB: 7P5J) and hTTYH3 (lower panel; PDB: 7P5C) in cartoon representation. The dashed rectangles frame zoom perspectives of the disulfide bridges within the ECDs. Cysteine residues are colored green and shown as spheres. **b** FSEC chromatograms depicting the elution profiles of DDM solubilized HEK 293 cells expressing the indicated EGFP-fused mTTYH1 (upper panel) and mTTYH3 (lower panel) before (gray curve) and following incubation with 20 mM 2-ME (colored curve). **c** Tetramer/dimer ratio of mTTYH1 (upper panel) and mTTYH3 (lower panel) before and following incubation with 2-ME. **d** Protein sequence alignment of the ECD region from mTTYH1 and mTTYH3, highlighting conserved positions and cysteine residues in green and yellow, respectively. **e** FSEC chromatograms depicting the elution profiles of mTTYH3 ECD cysteine mutants. **f** Tetramer/dimer ratio of the mTTYH3 ECD cysteine mutants. ($n = 3$, $t$-test, ns non-significant, ** $p < 0.01$).

chains (Fig. 3f, g). Importantly, we could not detect the emergence of a putative monomeric population at any insult temperature or detergent used for either mTTYH1 or mTTYH3. These results suggest that the tetramers dissociate into dimers, which are the obligatory indivisible building block of the mTTYH complexes, as observed in the cryo-EM structures[14,15]. However, these structures also exposed a surprising variability in the dimeric interface organization among the different hTTYH paralogs, resulting from rigid body rotation along the two-fold axis of the individual monomers[14], suggesting that the dimers are also adversely affected by detergents.

To gain further insight into the structural ramifications of detergent solubilization, we resorted to HDX-MS analyses of both dimeric and tetrameric mTTYH1 populations. Unexpectedly, mapping the HDX profile onto its homology model revealed an unforeseen pattern, with high deuterium uptake in interfacial regions and poorly exchanging peripheral regions (Fig. 4a). Intriguingly, an HDX-MS study of the human excitatory amino

acid transporter (EAAT) revealed that detergent solubilization significantly destabilizes the inter-subunit interface, leading to a complete loss of activity[45]. Moreover, structures of the dimeric UraA–proton symporter showed the direct interference of detergent molecules with inter-subunit interface formation, also resulting in non-functional transporters[46,47]. Thus, a similar mechanism may lead to destabilization of TTYH tetramerization interface following detergent solubilization, giving rise to stable dimers in solution and hampering the grasp of structure–function relations in this family.

Finally, we wondered if the tetramer is stabilized by the interaction between the ECDs, as observed in the cryo-EM structures of the dimers. These structures revealed that the ECDs are stabilized by two intra-subunit disulfide bridges (Fig. 5a). Moreover, HDX-MS analysis showed that the ECDs are relatively stabilized in the context of the tetramers (Fig. 4b, c). Hence, we examined the tetramer/dimer ratio following the incubation with the reducing agent 2-ME, showing a prominent shift toward dimeric stoichiometry

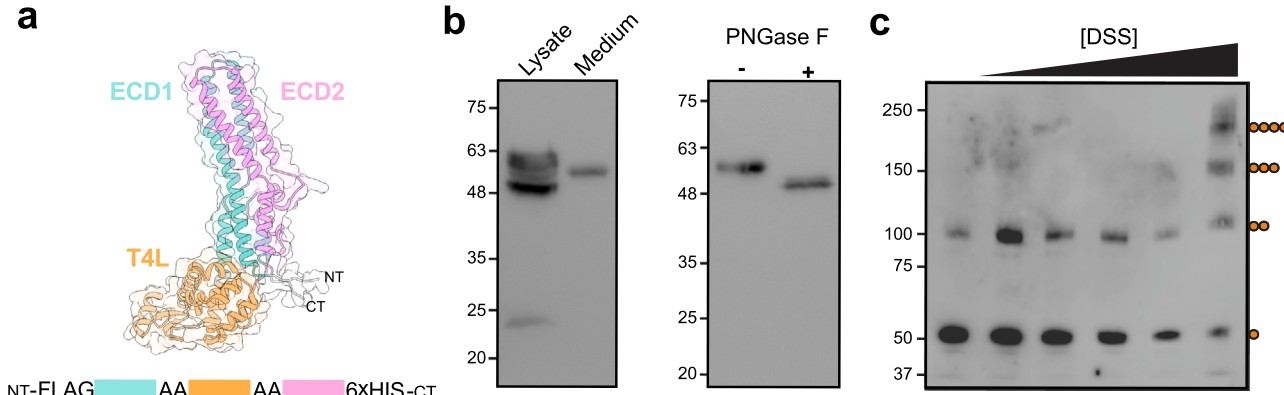

**Fig. 6 mTTYH3 ECDs form tetramers in solution. a** A model of ECD1-T4L-ECD2 was generated using AlphaFold 2[31]. **b** Western blot analyses of mTTYH3 ECD1-T4L-ECD2 expression using anti-FLAG. Note the single band observed in the secreted fraction (Medium), compared with the multiple intracellular forms (Lysate) (left panel), and the effect PNGase F treatment (500 units) (right panel). **c** Cross-linking using increasing concentrations of DSS (concentrations used were 0, 10, 50, 100, 200, and 2000 µM from left to right) resulted in the exposure of higher oligomeric assemblies, culminating in a mass corresponding to that expected for a tetramer.

(Fig. 5c). This observation supports the importance of intact disulfides for the tetramerization surfaces of mTTYH complexes. Indeed, targeting one of the ECD intra-subunit disulfide bridges by site-directed mutagenesis, also resulted in a low tetramer/dimer ratio (Fig. 5e, f). Moreover, cross-linking of isolated ECDs unveiled the formation of higher-order oligomers (Fig. 6c) consistent with tetramers, further supporting the stoichiometry observed in cellular membranes, and underscoring the role of these soluble domains in complex formation.

In conclusion, here we expose for the first time that mTTYH members form homotetramers at the plasma membrane using in situ cross-linking and single-molecule fluorescence microscopy. Our analysis suggests a dimer of dimers assembly mode, which involves the ECDs. Moreover, this tetrameric assembly is detergent-sensitive and is affected by the solubilization conditions. Importantly, our findings should guide future structural investigations of TTYH in the tetrameric context, which may illuminate the function of these enigmatic proteins.

## Materials and methods

**DNA constructs**. For FSEC, mTTYH1 and mTTYH3 were subcloned into the pEGFP-N1 vector (Clontech) using the Gibson assembly method[48]. For BN-PAGE and in situ cross-linking experiments, the same constructs were modified with the introduction of a FLAG tag instead of the original EGFP sequence. For subunit counting experiments, the intact mTTYH-EGFP fusions were subcloned into the pGH19 vector. For expression and purification of mTTYH1, intact mTTYH was subcloned into the pFastBac vector (Thermo Fisher) in-frame with a C-terminal EGFP and 8× His tag. For expression of the isolated mTTYH3 ECDs, the ECD1-T4L-ECD2 construct (Fig. 6a) was subcloned seamlessly between an N- and C-terminal FLAG and 6× His tag, respectively, into the pK503-9 vector. A list of primers used in this study is provided in Supplementary Table 1.

**Cross-linking analysis**. HEK 293 cells (HEK293T (ATCC; CLR-1573)) were transiently transfected with FLAG-tagged mTTYH constructs using calcium phosphate precipitation. Thirty-six hours post-transfection, cells were washed twice with PBS, followed by cross-linking of the adherent cells by incubation with BS³ (Thermo Fisher Scientific) at 10, 50, 100, 200, and 2000 µM for one hour at room temperature (RT). Subsequently, the reactions were quenched by the addition of Tris-HCl (pH 7.5) at a final concentration of 20 mM for 15 min. Cells were collected and centrifuged at 500×g for 3 min, resuspended with lysis buffer (150 mM NaCl, 50 mM HEPES, pH 7.5, 1% (w/v) DDM (Anatrace), 1 mM phenylmethylsulfonyl fluoride (PMSF), protease inhibitor cocktail set III (Merck Millipore)), incubated for 1 h at 4 °C and clarified by centrifugation at 21,000×g for 10 min. Finally, the supernatant was subjected to western blot analyses using mouse anti-FLAG (M2) (Sigma).

**Single-molecule subunit counting**. cRNAs were synthesized in vitro from linearized DNA of EGFP-fused mTTYH constructs using HiScribe T7 ARCA mRNA Kit (New England Biolabs). About 1 ng of cRNA were injected into Xenopus laevis

oocytes and imaged after 18 h of incubation at 18 °C. Imaging was performed using a total internal reflection fluorescence microscopy (TIRFM) system (Zeiss)[22]. Briefly, prior to the experiment, cells were manually devitellinized and placed on high refractive index coverslips with the animal pole in front of a Zeiss alpha Plan-Fluar 100X/1.45 NA oil immersion objective at RT. EGFP-fused mTTYH subunits were excited using an argon 488 (100 mW) laser and fluorescence was obtained using a 500 nm long-pass dichroic mirror in combination with a 525/50 nm band-pass filter. Images were collected at 30 Hz using a cooled electron-multiplying charge-coupled device camera (Photometrics). All the isolated and immobile diffraction-limited spots were subjected to bleaching steps analysis. The fluorescence intensity of each spot was plotted over time and the bleaching steps were manually counted. 338, 375, and 103 spots were counted from 2 to 6 oocytes using three different batches for mTTYH1, mTTYH3, and KCNH1.

**BN-PAGE**. HEK 293 cells were transiently transfected with FLAG-tagged mTTYH constructs using the calcium phosphate precipitation method. Forty-eight hours post-transfection, cells were washed twice in phosphate-buffered saline (PBS) and lysed with a buffer containing 50 mM NaCl, 20 mM Tris-HCl, pH = 7.5, 0.05% (for mTTYH1) or 0.1% (for mTTYH3) (w/v) DDM (Anatrace), protease inhibitor cocktail set III (Merck Millipore), 1 mM PMSF. Cell lysates were cleared by centrifugation (21,000×g for 30 min, 4 °C). Following determination of total protein concentration, equal quantities of protein were diluted 3:1 with native-PAGE sample buffer (50 mM Tris-Acetate, pH = 7, 6 M HCl, 50 mM NaCl, 10% (w/v) glycerol, 0.001% (w/v) ponceau S and 0.25% (w/v) Coomassie G), and were analyzed using 10% Tris-acetate gels and western blot with mouse anti-FLAG (M2) (Sigma).

**FSEC and FSEC-TS analyses**. HEK 293 cells were transiently transfected with the mTTYH-EGFP fusion constructs. After incubation for 48 h, cells were washed with PBS, harvested by gentle pipetting, and resuspended in 500 µl solubilization buffer (150 mM NaCl, 50 mM Tris-HCl, pH 7.5, 1% (w/v) DDM (Anatrace), protease inhibitor cocktail set III (Merck Millipore), 1 mM PMSF. Resuspended cells were rotated for 1 h at 4 °C, followed by centrifugation at 21,000×g for 1 h. About 50 µl of the supernatant were loaded onto a Superdex 200 Increase 10/300 GL column (Cytiva) preequilibrated with size-exclusion buffer (150 mM NaCl, 50 mM Tris-HCl, pH 7.5, 1 mM DDM (Anatrace)) in line with a fluorescence detector (RF-20A; Shimadzu Scientific Instruments), with λexcitation/λemission of 488/507 nm, respectively. For FSEC-TS, clarified samples were incubated at the indicated temperatures for 10 min using a thermal cycler and then centrifuged at 21,000×g for 10 min prior to the FSEC analysis described above.

**mTTYH1-GFP expression and protein purification**. Recombinant baculoviruses were generated using the Bac-to-Bac system (Thermo Fisher Scientific). The intact mTTYH1-GFP-8xHis tag fusion construct, subcloned into pFastBac, was transformed into DH10Bac E. coli strain to generate the recombinant bacmid DNA. Sf9 cells (Thermo Fisher Scientific; 11496015) were cultured at 27 °C with ESF-921 insect cell culture medium (Expression Systems) and transfected with bacmid DNA to generate P1 virus. P2 virus was generated by infecting cells at 1 × 10⁶ cells/mL with P1 virus (1:10, v/v) and harvested after 72 h. P3 and P4 virus stocks were generated in a similar manner (1:5, v/v), and infected cell pellets were flash-frozen in liquid nitrogen and stored at −80 °C. A cell pellet from 3.5 L culture was thawed on ice and lysed in 100 mL of buffer containing 50 mM Tris pH 7.5, 150 mM NaCl, 5 mM ethylenediaminetetraacetic acid (EDTA), 2 mM phenylmethylsulfonyl fluoride (PMSF), 1 mM 4-(2-aminoethyl) benzenesulfonyl fluoride (AEBSF), 4 µM

pepstatin A, 2.5 µg/mL aprotinin, 2 µg/mL leupeptin, 1 mM benzamidine. Cells were lysed using Emulsiflex C3 (Avestin), followed by initial clarification at 6500 × g for 10 min at 4 °C. Next, the membrane fraction was collected by centrifugation at 100,000×g for 1.5 h and homogenized with an electric disperser (IKA) in 30 mL lysis buffer containing 20 mM Tris pH 7.5, 150 mM NaCl, 2 mM PMSF, 1 mM AEBSF, 4 µM pepstatin A, 2.5 µg/mL aprotinin, 2 µg/mL leupeptin, 1 mM benzamidine, 1% $n$-dodecyl-β-D-maltopyranoside (DDM, Anatrace). Protein was extracted with gentle nutation for 1.5 h at 4 °C. Insoluble material was pelleted by centrifugation at 30,000×g for 30 min at 4 °C. The supernatant was incubated with preequilibrated TALON Metal Affinity Resin (Takara), following the addition of 8 mM imidazole, with gentle nutation overnight 4 °C. Next, the resin was collected in a column and washed with 7 column volumes of wash buffer (20 mM Tris, pH 7.5, 150 mM NaCl, 0.425 mM DDM) supplemented with 8 mM imidazole, followed by elution with 10 column volumes of wash buffer supplemented with 300 mM imidazole. About 1 mL fractions were collected and analyzed using SDS-PAGE. Fractions containing peak concentrations of purified protein were pooled and concentrated using a 50 kDa MWCO centrifuge concentrator to ~500 µL, and clarified by centrifugation at 21,000×g for 10 min at 4 °C. The concentrated protein was subjected to size-exclusion chromatography using a Superdex 200 Increase 10/300 column (Cytiva), preequilibrated with 20 mM Tris pH 7.5, 150 mM NaCl, and 0.3 mM DDM. Peak fractions were collected and spin concentrated for downstream experimentation.

**Size-exclusion chromatography multi-angle light scattering**. Experiments were performed using an analytical SEC column (Superdex 200 Increase 10/300 GL; Cytiva) preequilibrated with gel filtration buffer (20 mM Tris, pH 7.5, 150 mM NaCl, 0.3 mM DDM). Samples of purified mTTYH1-GFP-8xHis (50 µL, 1.18 mg/mL) were loaded onto the SEC column connected in line to a downstream 8-angle light scattering detector, followed by a differential refractive index detector (Wyatt Technology). Protein conjugate analysis was performed using a dedicated program within the Astra software package (Wyatt Technology) to determine the molecular mass of the protein and detergent components of the complex.

**Hydrogen/deuterium exchange mass spectrometry**. Hydrogen/deuterium exchange was started by a tenfold dilution of 20 µM protein solution into a D$_2$O-based buffer (20 mM Tris-Cl pH 7.5, 150 mM NaCl, 0.3 mM DDM). Aliquots were collected at different time points and quenched with 0.5 M glycine-HCl, pH 2.5, 0.5 M TCEP, 0.6 mM DDM in a 1:1 ratio and flash-frozen in liquid nitrogen. The labeling experiments were done at 25 and 4 °C. The 25 °C experiments were performed at four time points (20 s, 2 min, 20 min, and 2 h), with two time points (20 s and 20 min) replicated ($n = 3$). The 4 °C experiments were performed at four time points (2, 4, 6, 10, and 180 s), with the 2 and 6 s time points replicated ($n = 3$). The second set of HDX experiments was done as follows: each sample was thawed, left on ice for disulfide bond reduction for 3 min, and injected onto a liquid chromatography (LC) system consisting of immobilized pepsin/nepenthesin-2 custom-made column (bed volume 66 µL), trap VanGuard Pre-column (ACQUITY UPLC BEH C18, 130 Å, 1.7 µm, 2.1 mm × 5 mm, Waters) and an analytical column (ACQUITY UPLC BEH C18, 130 Å, 1.7 µm, 1 mm × 100 mm, Waters). The LC system was connected to an electron spray ionization (ESI) source of 15 FT-ICR MS (solariX XR, Bruker Daltonics). Digestion and peptide desalting with 0.4% formic acid (FA) in water was driven by a 1260 Infinity II Quaternary pump (Agilent Technologies) pumping at 100 µL min$^{-1}$ and took 3 min. Water-acetonitrile (ACN) gradient (5%–45% B within 7 min; solvent A: 0.1% FA in water, solvent B: 0.1% FA/2% water in ACN) followed by a quick step to 99% B (5 min) was used for elution and separation of the desalted peptides. The solvents were delivered by 1290 Infinity II LC System (Agilent Technologies) at 40 µL min$^{-1}$. To minimize the back-exchange, the setup was cooled to 0 °C. Acquired data were peak picked in Data Analysis (Bruker Daltonics) and exported to text files. These files were then loaded to DeutEx software[49] together with the sequence of the mTTYH1 construct and a list of identified peptides (see below). The processed HDX data were visualized in PyMol and using MSTools (http://peterslab.org/MSTools/index.php)[50], and a back-exchange correction was done[51,52]. Peptide identification for mTTYH1 dimeric and tetrameric populations was done using the same LC setup as described above but with the ESI-TIMS ToF Pro PASEF (Bruker Daltonics). Here 100 ms mobility ramps and 10 MS/MS events per a TIMS separation were used. Non-deuterated (ND) samples for these analyses were prepared similarly to deuterated ones, but the buffer was prepared using normal H$_2$O. To enable identification of the N-glycosylated sites, part of the proteins was deglycosylated by EndoH (New England Biolabs)—10 µg of mTTYH1 was incubated with 500 U of the enzyme at 30 °C overnight. LC-MS/MS data were searched against a custom-built database using the MASCOT search engine (v. 2.7, Matrix Science, London, United Kingdom). The database combined common cRAP.fasta (https://www.thegpm.org/crap/) with the sequence of mTTYH1, pepsin, and nepenthesin-2. Search parameters were as follows—variable modification HexNAc at Asn, precursor tolerance 10 ppm, fragment ion tolerance 0.05 Da. Decoy search was enabled, and FDR was set to <1% and IonScore >15. Glycopeptides were manually identified in the glycosylated ND data based on the presence of oxonium ions in the MS/MS spectra and the known identity of the peptide part derived from the analysis of deglycosylated samples. The type of glycan on each site and the distribution of glycoforms were identified. Most intense glycoforms were then included in the input file for DeuteEx software.

**Isolated mTTYH3 ECDs expression in insect cells**. Recombinant baculoviruses were generated using the Bac-to-Bac system (Thermo Fisher Scientific). mTTYH3 ECD1-T4L-ECD2 was cloned into pK503-9, a pFastBac derivative, to generate a fusion construct with N-terminal FLAG tag and C-terminal 6xHis tag. The plasmid was transformed into DH10Bac $E. coli$ strain to generate the recombinant bacmid DNA. Sf9 cells were cultured at 27 °C with ESF-921 insect cell culture medium (Expression Systems) and transfected with bacmid DNA to generate the P1 virus. P2 virus was generated by infecting cells at 1×10$^6$ cells/mL with P1 virus (1:10, v/v) and harvested after 72 h. P3 and P4 virus stocks were generated in a similar manner (1:5, v/v). The media containing the secreted protein was clarified by centrifugation at 1200×g for 5 min.

**Homology modeling**. The structure of a single mTTYH1 chain was modeled using AlphaFold 2[31] (Identifier AF-Q9D3A9-F1). The dimer was assembled by superposition of two monomers onto the structure of hTTYH1 (PDB 7P5J).

**Statistics and reproducibility**. Statistical analyses were carried out in GraphPad Prism version 9 (GraphPad Software). Two-group comparisons were performed using an unpaired $t$-test assuming Gaussian distribution. No statistical methods were used to predetermine sample sizes. Required experimental sample sizes were chosen according to common practice in protein biochemistry and microscopy (at least three independent experiments). Statistical analysis was limited to determining mean ± SEM.

**Reporting summary**. Further information on research design is available in the Nature Research Reporting Summary linked to this article.

## Data availability

The mass spectrometry proteomics data have been deposited to the ProteomeXchange Consortium via the PRIDE partner repository with the dataset identifier PXD031833. Uncropped gel images corresponding to Figs. 1a–c, 3a, 6b, 6c, and Supplementary Fig. 1 are available in Supplementary Data 2. Any other relevant data are available from corresponding authors upon request.

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

## Acknowledgements

This work was performed in partial fulfillment of the requirements for a Ph.D. degree of E.M., Sackler Faculty of Medicine, Tel-Aviv University, Israel. This work was supported by the Israel Science Foundation (grants 1721/16 and 1653/21) (Y.H.), the Israel Cancer Research Fund grants 01214 (Y.H.) and 19202 (M.G.), and the German-Israeli Foundation for Scientific Research and Development (grant No. I-2425-418.13/2016) (Y.H.). Support also came from the I-CORE Program of the Planning and Budgeting Committee and the Israel Science Foundation (grant 1775/12) (Y.H.), Recanati Foundation (M.G.), Marguerite Stolz Research Fellowship (Y.H.), Kahn foundation's Orion project, Tel-Aviv Medical Center, Israel (M.G.), Israel Cancer Association (grant 20200037) (Y.H. and M.G.), and the Claire and Amedee Maratier Institute for the Study of Blindness and Visual Disorders, Sackler Faculty of Medicine, Tel-Aviv University (Y.H. and M.G.). Access to the EU_FT–ICR_MS network installation was funded by the EU Horizon 2020 grant 731077. Support of the BioCeV center (CZ.1.05/1.1.00/02.0109) and the CMS/CIISB facility supported by MEYS CR (LM2018127) is also gratefully acknowledged.

## Author contributions

Conceptualization, M.G. and Y.H.; Methodology, M.G., P.M., and Y.H.; Investigation, E.M., E.S., and Z.K.; Formal analysis, E.M., E.S., Z.K., P.M., M.G., and Y.H.; Writing—original draft, E.M., M.G., and Y.H.; Writing—review and editing, E.M., Z.K., P.M., M.G., and Y.H.; Supervision, M.G., P.M., and Y.H.; Funding acquisition, M.G. and Y.H.

## Competing interests

The authors declare no competing interests.
