## [Peer Review File · Communications Biology]

Reviewers' comments:

Reviewer #1 (Remarks to the Author):

This manuscript by Haitin group attempts to demonstrate that TTYH proteins form tetramer. The authors report a series of evidence, including in situ cross-linking analysis and single-molecule fluorescence counting, and arrive at the conclusion that mTTYH1 and mTTYH3 form tetramers at the plasma membrane, stabilized by interactions between extracellular domains. The main challenge in membrane protein purification is the determination and optimization of the chemical conditions capable of solubilizing the protein from the membrane and stabilizing its native state. The denaturation of the membrane protein after detergent solubilization may interfere with understanding of structure-function relations. Thus, this study is a very important piece of work to understand the structure of mTTYH members virtually resembling their native state. This is in stark disagreement with the recent cryoEM results proposing a dimer stoichiometry, whereas this is in agreement with a prediction made by Han et al., 2019, *Experimental Neurobiology*, which should be mentioned in the introduction or discussion. This study provides convincing series of evidence to support their claim. The techniques the authors employed are appropriate and cutting-edge. Authors make adequate conclusions about their results. This reviewer highly recommends this paper for publication at this journal. However, there are a few minor comments that need to be addressed for a minor revision.

1. Based on their findings, it would be very helpful for the field if authors could generate a tetrameric form of TTYH using the currently available homodimeric TTYH from the cryoEM studies.

2. To prove the structure of mTTYH1 and 3 are tetramer, the authors used other tetrameric channel as a positive control. For example, KCNH1 was used as a positive control as a known tetrameric channel by single-molecule subunit counting. To obtain more convincing evidence for tetrameric form of the mTTYH members but not for homodimer as previously published by cryo-EM (Sukalskaia A et al., *Nat. Common* (2021), Li B., *Nat. Common* (2021)), it would be informative to include a negative control as a known homo-dimeric channel.

3. In the preprint paper posted in bioRxiv, the same authors showed that the tetramer/dimer ratio was increased by DDM-CHS treatment. However, in this submitted manuscript, the authors only demonstrated a decreased tetramer/dimer ratio after 2ME treatment in DDM-CHS condition without any explanation for the rationale of using CHS treatment. It would be helpful to provide an accurate rationale for using specific chemicals in these experiments.

4. In the very first sentence of the Results section, "To estimate the stoichiometry of TTYH proteins within their native membrane environment, we overexpressed FLAG-tagged mTTYH paralogs in HEK 293 cells (Fig. 1a)." This sentence is missing "and performed western blot analysis."

Reviewer #2 (Remarks to the Author):

In the manuscript 'TTYH family members form tetrameric complexes at the cell membrane' by Melvin et al. the authors claim that the TTYH family receptors form homo-tetramers in the native cell environment rather than dimers which was the predominant opinion so far. They used different methods to determine the stoichiometry, i.e. crosslinking and SDS-PAGE western blot, single-molecule subunit counting, STORM and HDX-MS. It is remarkable that the authors use several different approaches to demonstrate that the previously observed dimers can indeed be generated under different conditions, e.g. the use of detergents, which destabilizes the tetrameric structures. They find multiple regions that stabilize the tetrameric structure, e.g. the exposed interfacial regions in extracellular domain and the intramolecular disulphide. Finally, they show that the ECD of TTYH 3 alone can form tetramers. This study is an important contribution to the ongoing research about TTYH

family receptors structure and function. There are few concerns regarding the explanations and conclusions that should be addressed before publication, but I don't think that additional experiments are required.

Major issues:

1. The clustering analysis of Fig. 3c and 3f does not seem valid to me. Probably any protein (also a monomeric protein) will give a similar radius, since this is probably the distribution of localizations in the imaging setup. In the cited literature, a clustering analysis was done for differentially labeled proteins, and a peak at 50nm for the distance between two different colors was used to claim that they are in a complex of 50nm or less. However, this is very different from making claims about the actual complex size. To achieve support for a certain complex size, it would be necessary to calibrate the approach with a known protein of very small complex size (e.g. 5nm), to get the response function of the instrument and method. In addition, in Fig. 3a it is quite obvious that drift of the stage caused an elongation of the clusters in the vertical direction. Accordingly, the approach does not at all give support for a tetrameric structure.
2. The BN-PAGE western blot analysis of DDM solubilized mTTYH1 in Fig. 4a shows a significant amount of the tetrameric compared to the dimeric population. The reason behind this significant presence of tetramers even after detergent solubilization is unclear. Also, the authors have argued that the dimeric population may arise from the use of Coomassie stain in BN-PAGE, but what can be the possible compound in the stain and by what mechanism is it causing the tetramers disassembling to dimers?
3. DDM is used to see the effect of detergents on TTYH tetramers. While for BN-PAGE, TTYH1 tetramers remain intact and TTYH3 tetramers decay to dimers, for FSEC, the TTYH3 tetramers are much more stable, and TTYH1 tetramers almost disappear. The reason behind this skewness should be discussed.
4. It is stated that there is an effect of charged groups on the disassembly process of tetramers. The authors have argued that the effect of LDAO (a detergent with the same alkyl chain length as DDM but with an amine oxide headgroup) on TTYH paralogs shows the FSEC elution profile altered and that the detergent increases the disassembly of tetramers to dimers. But the result shown in Fig. 4e and 4f, i.e. the elution profile for mTTYH1 and mTTYH3 after treatment with both DDM and LDAO, is quite different. DDM alone causes a significant increase in the dimeric population in case of mTTYH1, even slightly more than LDAO. And while DDM is unable to destabilize tetramers of mTTYH3, the elution profile of mTTYH3 oligomers after treatment with LDAO is still skewed more towards tetramers than towards dimers. This part should be better explained.

Minor issues:

1. In the images of the TTYH spots in Fig. 2a and 2d, there are many bright clusters. Are these caused by TTYH? Also, the authors should state whether ALL immobile/separated spots were chosen for analysis (in Fig. 1a, there is a large number of spots that were not selected for analysis). A movie of the images should be presented. In the text, the authors write that the baseline was reached after 100-200 s. Do they mean 10-20 s?
2. Supp. Fig 1a suggests that with a probability p even smaller than 0.7, the fit would even be better.
3. The concentrations of BS3 in Fig. 1 should be explicitly stated, either in the figure or in the Methods section. It only says 10-2000 μ M.
4. Fig. 5a, b: a cross-section through the middle and view from top would be helpful to see what helices are more or less accessible.

Reviewer #3 (Remarks to the Author):

In this manuscript, the authors studied the stoichiometry of 2 mouse Tweety homolog (TTYH) family proteins, mTTYH1 and mTTYH3. Previous studies of human TTYH structures had shown that these proteins exist as dimers. In this manuscript, however, the authors demonstrated that mTTYH1 and mTTYH3 are likely to form tetramers instead of dimers on the cell surface using Western and single-

molecule fluorescence microscopy. In addition, the authors went on to investigate the reason for the discrepancy between their study and previous structural studies and concluded that the detergent used to solubilize the protein leads to the dissociation of the tetramers. Finally, the authors identified that the ECDs of mTTYHs seem to aid tetramerization.

Although the TRIFM subunit counting experiment is very appropriate and sound, some of the other experiments are not as well designed and do not convincingly support the main idea of the paper.

Specifically,

1. In Figure 1, because the crosslinker is not membrane-permeable, the cross-linked subunits are assumed to be on the plasma membrane. However, the authors did not include any controls to show the integrity of the cell membrane. If the membrane becomes leaky due to the transfection, the crosslinker can get into the cells and crosslink the intracellular mTTYHs. And as Figure 4 shows, in the whole cell lysate, most of the mTTYH proteins are assembled as tetramer even without cross-linking. Thus, the small amount of the tetramer seen in Figure 1 (especially in Fig.1c) are not necessarily from the plasma membrane.

2. It is important for the authors to provide un-cropped and un-processed western films as source data.

3. The STORM experiment and analysis clearly show that the mTTYHs are multimers on the cell surface. The reasoning the authors used to conclude that mTTYHs exist as tetramers is that their cluster radii are similar to that of KCNQ channels. However, the authors themselves did not perform the same experiment using KCNQ. Instead, they cited a paper that expressed KCNQ channels in CHO cells instead of HEK293 cells. It is important for the authors to perform such experiment using their own system to see whether different cell lines affect the results. In addition, if possible, the authors should consider including a negative control: use a protein with similar size as mTTYH that is also known to exist as a dimer on the cell membrane and perform the same analysis.

4. Have the authors run the peaks in Fig 4b using BN-PAGE? It is important to note that the authors did not purify the TTYH proteins in this experiment, so the different peaks seen in the FSEC might be some complexes formed by the TTYH and other unknown proteins. Thus, it is important to, at the bare-minimum, analyze the peaks using BN-PAGE and Western. From the FSEC profile, the peak assumed to be the dimer of TTYH3 is about 1 ml earlier than the peak assumed to be the dimer of TTYH1. However, from Fig 4a, the size difference between dimer-TTYH1 and dimer-TTYH3 is less than 30kD. The elution volume difference seems to be too large. In addition, if the authors analyze the peaks assumed to be TTYH tetramers using BN-PAGE and Western and only see a tetramer band, it will rule out the possibility that "the dimeric population may arise from the reagents used for the BN-PAGE analysis".

5. Because it had not been convincingly established that the peaks in Fig 4b correspond to dimeric and tetrameric TTYHs, Figure 5 and Figure 6 become less meaningful as well. For the HDX-MS experiment, the authors should include LC-MS quality control and a list of all the peptides/proteins identified from the mass spec experiment from the MASCOT search including peptide count, peptide score, p-values, etc. as supplementary data. How many times have the authors done the HDX-MS experiment at 4 degrees? Only one result is shown in the paper. I believe at least 1 more replication needs to be included as supplementary data for the result to be convincing.

6. From the HDX-MS result, the authors identified a disulfide bond (C303-C370) that could contribute to tetramerization. However, by treating the protein with 2-ME, all disulfide bonds in all proteins are reduced. Thus, one cannot conclude that the assumed tetramer-to-dimer shift is due to the break of "intra-subunit disulfides" as the subtitle suggests. A more appropriate experiment is to mutate either C303 or C370 to alanine to only break that one disulfide bond and test whether there is a shift towards the assumed dimer peak.

7. Could the authors explain in more detail why they express the engineered mTTYH3 ECDs as a secreted protein using insect cells and why they cross-link the secreted ECDs? I believe that the point of this study is to prove that in native conditions, the TTYHs form tetramers on cell membrane. The experiment in Figure 7 is neither native nor on the cell membrane. I don't believe it is necessary and particularly meaningful.

Reviewer #4 (Remarks to the Author):

The manuscript authored by Melvin et al tries to demonstrate the tetrameric organization of Tweety homolog (TTYH) family proteins in native cellular membrane where they are potentially display and form Cl⁻ conducting channels in the membranes. The authors employ independent methodologies of crosslinking, superresolution microscopy, FSEC and HDX-MS to demonstrate the oligomerization in the case of TTYH family proteins. They generally conclude that detergent extraction destabilizes the tetramer and leads to dimeric arrangement as visualized in recent cryoEM structures. I have the following major concerns about this study.

1. While the authors seem to have done a thorough analysis of TTYH clustering using dSTORM analysis and do obtain a distribution that indicates tetramers of TTYH in the membrane, it was not clear how the bleach step assignment was done in Fig. 2e. The steps in this are not very clear.

2. Given the expertise of the authors in HDX-MS based on the deuterium exchange pattern, can the authors model a tetrameric form of the TTYH1 and TTYH3 in a way that can satisfy the contact analyses.

3. Related to the above, the notion of disulfides mediating tetramer formation is hTTYH1 in figure 6. One would have expected the authors to use a detergent system that allows formation of a tetramer in case of TTYH1, like LMNG, to perform the 2-mercaptoethanol exposure. Using DDM/CHS does not help tetramer formation (dashed line) in this case and one cannot conclude clear tetramer/dimer formation. This experiments clearly needs to be redone.

4. Related to the above, it would have been interesting to see how the profile shifts with the inclusion of different lipids including CHS, POPC, native lipid extracts to see effect on oligomerization. Contrary to the findings in this manuscript, the cryoEM structures reported by the Dutzler group determines the dimeric structures of the TTYH molecule within nanodiscs and GDN. One would presume that the TTYH oligomers in nanodiscs would be in the native tetrameric form if this is accurate.

5. Can the authors demonstrate the presence of any Cl⁻ conductance in HEK cells expressing the TTYH orthologues studied in this manuscript.

6. Given that LMNG and DMNG extract tetrameric forms of the molecule, can the authors demonstrate negative stain EM or reference free 2D classes in cryoEM to validate their claim. This would provide a very clear proof of their findings.

Minor concerns.

Page 3 line 1. Represent scientific names accurately. *Drosophila melanogaster* and use capital D for *drosophila*.

Reviewer #1:

This manuscript by Haitin group attempts to demonstrate that TTYH proteins form tetramer. The authors report a series of evidence, including in situ cross-linking analysis and single-molecule fluorescence counting, and arrive at the conclusion that mTTYH1 and mTTYH3 form tetramers at the plasma membrane, stabilized by interactions between extracellular domains. The main challenge in membrane protein purification is the determination and optimization of the chemical conditions capable of solubilizing the protein from the membrane and stabilizing its native state. The denaturation of the membrane protein after detergent solubilization may interfere with understanding of structure-function relations. Thus, this study is a very important piece of work to understand the structure of mTTYH members virtually resembling their native state. This is in stark disagreement with the recent cryoEM results proposing a dimer stoichiometry, whereas this is in agreement with a prediction made by Han et al., 2019, *Experimental Neurobiology*, which should be mentioned in the introduction or discussion. This study provides convincing series of evidence to support their claim. The techniques the authors employed are appropriate and cutting-edge. Authors make adequate conclusions about their results. This reviewer highly recommends this paper for publication at this journal. However, there are a few minor comments that need to be addressed for a minor revision.

Response: We thank the reviewer for the constructive and judicious comments and advice. The manuscript was modified in accordance with the Reviewer's constructive critique. As suggested, we referenced the stoichiometry prediction made by Han et al., 2019:

"This is in contrast to the predicted tetrameric organization inferred from electrophysiological studies." (Page 3, line 50).

Additionally, our point-by-point responses are appended below.

1. Based on their findings, it would be very helpful for the field if authors could generate a tetrameric form of TTYH using the currently available homodimeric TTYH from the cryoEM studies.

Response: We thank the reviewer for raising this important point. Our data clearly support the tetrameric stoichiometry of TTYH proteins in their native environment. However, unfortunately, we cannot derive the high-resolution spatial information needed to generate a tetrameric model, as suggested. Specifically, the HDX-MS data provide residue-level backbone exchange information without any spatial data. Moreover, as our HDX-MS analysis indicates that detergent solubilization may adversely affect the conformation of TTYH subunits, using the dimeric structures as a starting point for further modeling of the tetrameric organization may result in highly inaccurate models. Therefore, while we agree that obtaining the tetrameric structure is highly important for the field, this problem should be addressed experimentally in the future.

2. To prove the structure of mTTYH1 and 3 are tetramer, the authors used other tetrameric channel as a positive control. For example, KCNH1 was used as a positive control as a known tetrameric channel by single-molecule subunit counting. To obtain more convincing evidence for tetrameric form of the mTTYH members but not for

homodimer as previously published by cryo-EM (Sukalskaia A et al., Nat. Commun (2021), Li B., Nat. Commun (2021)), it would be informative to include a negative control as a known homo-dimeric channel.

Response: Thank you for this important notion. Performing the subunit counting experiments, we relied on a well-developed methodology. Specifically, the Isacoff group previously determined the expected bleaching step distribution for different oligomeric states (Ulbrich and Isacoff, Nat. Methods, 2007). Notably, it is extremely rare to observe more than two bleaching steps in a dimeric assembly (< 5%; see also Tombola et al., Neuron, 2008), while in the case of TTYH, three to four bleaching steps are most frequently observed. Therefore, our data inherently refute the possibility of dimeric organization at the cell membrane. Finally, we would like to underscore that the tetrameric stoichiometry suggested here is supported by orthogonal and complementary approaches, and not solely on subunit counting. Specifically, to provide additional support to our suggested tetrameric stoichiometry, we performed SEC-MALS analysis of purified mTTYH1, revealing a protein mass of 223 and 109 kDa for the tetrameric and dimeric populations, respectively. The calculated masses are within that expected, considering the experimental error of the MALS measurement (revised Fig. 3c).

“Next, to validate the oligomeric state of TTYH in each FSEC elution peak, we proceeded with purification of intact mTTYH1-GFP for SEC-MALS analysis (Fig. 3c). Indeed, the peaks eluting at the volumes assumed to represent tetramers and dimers revealed a protein mass of 223 ± 0.4 and 109 ± 1.4 kDa for the tetrameric and dimeric populations, respectively. As expected, the mass of the tetrameric population is double that of the dimeric one, and both are within the experimental error of the MALS measurement.” (page 6, line 123).

3. In the preprint paper posted in bioRxiv, the same authors showed that the tetramer/dimer ratio was increased by DDM-CHS treatment. However, in this submitted manuscript, the authors only demonstrated a decreased tetramer/dimer ratio after 2ME treatment in DDM-CHS condition without any explanation for the rationale of using CHS treatment. It would be helpful to provide an accurate rationale for using specific chemicals in these experiments.

Response: As indicated in the manuscript text (page 9, line 195), Fig. 5 shows FSEC analyses of DDM-solubilized cells, in the presence or absence of 2-ME. We have corrected the figure legend in the revised version and removed the erroneous indication of CHS treatment. Thank you for bringing this important point to our attention. We further reinforced the significance of disulfide bridges for the structural integrity of TTYH tetramers by mutagenesis of mTTYH3 (Fig. 5d-f), as well as by showing a similar effect in LMNG-solubilized preparations (new Supplementary Fig. 6). As in the case of DDM-mediated solubilization, 2-ME resulted in tetramer dissociation into dimers, when the cells were lysed with LMNG, despite the initial higher tetramer/dimer ratio.

“Nevertheless, a similar effect was observed following solubilization with LMNG, which enriches the tetrameric population (Fig. 3f, g; Supplementary Fig. 6). To further pinpoint the disulfide bridge contributing to tetramer stabilization, we proceeded with analyses of mTTYH3-GFP ECD cysteine mutants' impact on tetramerization (Fig. 5d). Akin to the effect of 2-ME, the FSEC elution profile of C299A and C366A (Fig. 5e) resulted in a marked shift towards dimers (Fig. 5f).” (page 9, line 198).

“...Indeed, targeting one of the ECD intra-subunit disulfide bridges by site-directed mutagenesis, also resulted in a low tetramer/dimer ratio (Fig. 5e, f).” (page 13, line 288).

4. In the very first sentence of the Results section, "To estimate the stoichiometry of TTYH proteins within their native membrane environment, we overexpressed FLAG-tagged mTTYH paralogs in HEK 293 cells (Fig. 1a)." This sentence is missing "and performed western blot analysis."

Response: Added – thank you.

Reviewer #2:

In the manuscript 'TTYH family members form tetrameric complexes at the cell membrane' by Melvin et al. the authors claim that the TTYH family receptors form homo-tetramers in the native cell environment rather than dimers which was the predominant opinion so far. They used different methods to determine the stoichiometry, i.e. crosslinking and SDS-PAGE western blot, single-molecule subunit counting, STORM and HDX-MS. It is remarkable that the authors use several different approaches to demonstrate that the previously observed dimers can indeed be generated under different conditions, e.g. the use of detergents, which destabilizes the tetrameric structures. They find multiple regions that stabilize the tetrameric structure, e.g. the exposed interfacial regions in extracellular domain and the intramolecular disulphide. Finally, they show that the ECD of TTYH 3 alone can form tetramers. This study is an important contribution to the ongoing research about TTYH family receptors structure and function. There are few concerns regarding the explanations and conclusions that should be addressed before publication, but I don't think that additional experiments are required.

Response: We thank reviewer #2 for this overview of the manuscript, the thorough assessment, and insightful comments. In line with the reviewer's critique, we made significant changes to the text and added new experimental data, as detailed in our response below.

Major issues:

1. The clustering analysis of Fig. 3c and 3f does not seem valid to me. Probably any protein (also a monomeric protein) will give a similar radius, since this is probably the distribution of localizations in the imaging setup. In the cited literature, a clustering analysis was done for differentially labeled proteins, and a peak at 50nm for the distance between two different colors was used to claim that they are in a complex of 50nm or less. However, this is very different from making claims about the actual complex size. To achieve support for a certain complex size, it would be necessary to calibrate the approach with a known protein of very small complex size (e.g. 5nm), to get the response function of the instrument and method. In addition, in Fig. 3a it is quite obvious that drift of the stage caused an elongation of the clusters in the vertical direction. Accordingly, the approach does not at all give support for a tetrameric structure.

Response: We thank the reviewer for raising this important point! Indeed, following this criticism, we performed the STORM analysis using the dimeric voltage-gated proton channel Hv1. In line with your comment, Hv1 exhibited a similar cluster radius size distribution, indicating the lack of resolution to determine the oligomeric state of TTYH proteins. Therefore, we omitted the STORM section from the revised manuscript. Instead, to provide additional support to our suggested tetrameric stoichiometry, we performed SEC-MALS analysis of purified mTTYH1, revealing a protein mass of 223 and 109 kDa for the tetrameric and dimeric populations, respectively. The calculated masses are within that expected, considering the experimental error of the MALS measurement (revised Fig. 3c, appended on page 5).

“Next, to validate the oligomeric state of TTYH in each FSEC elution peak, we proceeded with purification of intact mTTYH1-GFP for SEC-MALS analysis (Fig. 3c). Indeed, the peaks eluting at the volumes assumed to represent tetramers and dimers revealed a protein mass of 223 ± 0.4 and 109 ± 1.4 kDa for the tetrameric

and dimeric populations, respectively. As expected, the mass of the tetrameric population is double that of the dimeric one, and both are within the experimental error of the MALS measurement.” (page 6, line 123).

Revised Fig. 3

2. The BN-PAGE western blot analysis of DDM solubilized mTTYH1 in Fig. 4a shows a significant amount of the tetrameric compared to the dimeric population. The reason behind this significant presence of tetramers even after detergent solubilization is unclear. Also, the authors have argued that the dimeric population may arise from the use of Coomassie stain in BN-PAGE, but what can be the possible compound in the stain and by what mechanism is it causing the tetramers disassembling to dimers?

Response: Thank you for this comment. The reason for the apparent discrepancy in the tetramer/dimer ratio stems from the DDM concentrations used in each experiment. As can be seen in the attached figure, the DDM concentration greatly impacts tetramers dissociation observed in BN-PAGE analysis. Indeed, the DDM concentration used in Fig. 4a (revised Fig. 3a, appended on page 5) was lower than that used in Fig. 4b (revised

Fig. 3b, appended on page 5). Notably, in either case, both tetrameric and dimeric populations are detected and this experiment was not used for quantitative assessment of the different species. As for the possible effect of the Coomassie stain, it may lead to the dissociation of fragile protein complexes due to a repulsive effect of superficial negatively charged residues (e.g. see Eubel et al., *Plant Methods*, 2005; reference 28 added to the revised manuscript).

3. DDM is used to see the effect of detergents on TTYH tetramers. While for BN-PAGE, TTYH1 tetramers remain intact and TTYH3 tetramers decay to dimers, for FSEC, the TTYH3 tetramers are much more stable, and TTYH1 tetramers almost disappear. The reason behind this skewness should be discussed.

Response: Thank you for pointing that out. As stated in our response to comment #2, these differences arise from the different detergent concentrations used. Moreover, as can be seen in the attached figure, mTTYH3 tetramers are indeed more resistant to increasing DDM concentrations, in accordance with our FSEC analyses. Please note that the BN-PAGE analysis is qualitative and is not interpreted quantitatively in the manuscript.

4. It is stated that there is an effect of charged groups on the disassembly process of tetramers. The authors have argued that the effect of LDAO (a detergent with the same alkyl chain length as DDM but with an amine oxide headgroup) on TTYH paralogs shows the FSEC elution profile altered and that the detergent increases the disassembly of tetramers to dimers. But the result shown in Fig. 4e and 4f, i.e. the elution profile for mTTYH1 and mTTYH3 after treatment with both DDM and LDAO, is quite different. DDM alone causes a significant increase in the dimeric population in case of mTTYH1, even slightly more than LDAO. And while DDM is unable to destabilize tetramers of mTTYH3, the elution profile of mTTYH3 oligomers after treatment with LDAO is still skewed more towards tetramers than towards dimers. This part should be better explained.

Response: Thank you for this important comment. We modified the results section to better reflect our experimental data, in line with your comment:

“Specifically, following solubilization of the cells with lauryl dimethylamine-N-oxide (LDAO), a detergent sharing an identical alkyl chain length with DDM, but containing a charged amine oxide headgroup, mTTYH1 displayed an elution profile reminiscent of that obtained with DDM, while mTTYH3 was partially skewed towards dimers. Thus, the use of a charged headgroup did not significantly affect the tetramer to dimer ratio.” (page 7, line 141).

Minor issues:

1. In the images of the TTYH spots in Fig. 2a and 2d, there are many bright clusters. Are these caused by TTYH? Also, the authors should state whether ALL immobile/separated spots were chosen for analysis (in Fig. 1a, there is a large number of spots that were not selected for analysis). A movie of the images should be presented. In the text, the authors write that the baseline was reached after 100-200 s. Do they mean 10-20 s?

Response: Yes, we assume that the bright clusters are caused by TTYH. All the spots that met the criteria described under the Materials and Methods section were included, as now explicitly stated under this section.

As requested, a representative movie (Supplementary Movie 1) is now available as Supplementary Material. We indeed meant 10-20 s, now corrected in the text – thank you.

2. Supp. Fig 1a suggests that with a probability p even smaller than 0.7, the fit would even be better.

Response: According to the reviewer's suggestion, we further tested lower values of the probability p (see attached figure; replaces Supplementary Fig. 2a). While lower values improved the fit in the 1-2 bleaching steps range, the agreement with the data was sub-optimal at higher bleaching steps. Therefore, we conclude that a $p = 0.7$ provides the overall best fit.

3. The concentrations of BS3 in Fig. 1 should be explicitly stated, either in the figure or in the Methods section. It only says 10-2000 μM .

Response: Thank you. BS³ concentrations are now stated in Fig. 1 legend and the relevant Methods section:

“Cross-linking analysis – HEK 293 cells were transiently transfected with FLAG-tagged mTTYH constructs using calcium phosphate precipitation. 36 hours post-transfection, cells were washed twice with PBS, followed by cross-linking of the adherent cells by incubation with BS³ (Thermo Fisher Scientific) at 10, 50, 100, 200, and 2000 μM for one hour at room temperature (RT).”... (Page 14, line 310).

4. Fig. 5a, b: a cross-section through the middle and view from top would be helpful to see what helices are more or less accessible.

Response: Thank you for this suggestion. Top view panels from the extracellular side have been added to Fig. 5 (revised Fig. 4).

Revised Fig. 4

Reviewer #3:

In this manuscript, the authors studied the stoichiometry of 2 mouse Tweety homolog (TTYH) family proteins, mTTYH1 and mTTYH3. Previous studies of human TTYH structures had shown that these proteins exist as dimers. In this manuscript, however, the authors demonstrated that mTTYH1 and mTTYH3 are likely to form tetramers instead of dimers on the cell surface using Western and single-molecule fluorescence microscopy. In addition, the authors went on to investigate the reason for the discrepancy between their study and previous structural studies and concluded that the detergent used to solubilize the protein leads to the dissociation of the tetramers. Finally, the authors identified that the ECDs of mTTYHs seem to aid tetramerization.

Although the TRIFM subunit counting experiment is very appropriate and sound, some of the other experiments are not as well designed and do not convincingly support the main idea of the paper.

Response: We appreciate the insightful comments provided by Reviewer #3. The manuscript was changed in accordance with the Reviewer's suggestions and new data were added to support our findings as specified below in our detailed response.

Specifically,

1. In Figure 1, because the crosslinker is not membrane-permeable, the cross-linked subunits are assumed to be on the plasma membrane. However, the authors did not include any controls to show the integrity of the cell membrane. If the membrane becomes leaky due to the transfection, the crosslinker can get into the cells and crosslink the intracellular mTTYHs. And as Figure 4 shows, in the whole cell lysate, most of the mTTYH proteins are assembled as tetramer even without cross-linking. Thus, the small amount of the tetramer seen in Figure 1 (especially in Fig.1c) are not necessarily from the plasma membrane.

Response: Thank you for raising this point. Indeed, validation of membrane integrity is vital for removing any doubts associated with *in situ* cross-linking. Therefore, to alleviate this concern, we used 14-3-3 γ as a control. 14-3-3 is a family of dimeric intracellular and globular proteins. As can be seen in the new Supplementary Fig. 1, BS³ failed to cross-link 14-3-3 γ while treatment of the cells with the membrane-permeable cross-linker disuccinimidyl suberate (DSS) yielded dimers. This supports our experimental design and further reinforces the tetrameric stoichiometry of mTTYH proteins observed using *in situ* cross-linking with BS³.

2. It is important for the authors to provide un-cropped and un-processed western films as source data.

Response: All un-cropped and un-processed western blots are now available as source data.

3. The STORM experiment and analysis clearly show that the mTTYHs are multimers on the cell surface. The reasoning the authors used to conclude that mTTYHs exists as tetramers is that their cluster radii are similar to that of KCNQ channels. However, the authors themselves did not perform the same experiment using KCNQ. Instead, they cited a paper that expressed KCNQ channels in CHO cells instead of HEK293 cells. It is important for the authors to perform such experiment using their own system to see whether different cell lines affect the results. In

addition, if possible, the authors should consider including a negative control: use a protein with similar size as mTTYH that is also known to exist as a dimer on the cell membrane and perform the same analysis.

Response: Thank you for this comment. Importantly, a similar concern was raised by Reviewer #2. Following further experimentation, we concluded that the cluster radii of tetrameric and dimeric proteins cannot be reliably discerned using this approach, and we decided to omit these results from the revised manuscript. Instead, to further support the suggested tetrameric stoichiometry of mTTYH proteins, we analyzed purified mTTYH1 using SEC-MALS. Our results, now presented in revised Fig. 3c (appended on page 5), show that the protein mass of mTTYH1 is 223 kDa, corresponding to a tetramer within the experimental error of the MALS measurement.

“Next, to validate the oligomeric state of TTYH in each FSEC elution peak, we proceeded with purification of intact mTTYH1-GFP for SEC-MALS analysis (Fig. 3c). Indeed, the peaks eluting at the volumes assumed to represent tetramers and dimers revealed a protein mass of 223 ± 0.4 and 109 ± 1.4 kDa for the tetrameric and dimeric populations, respectively. As expected, the mass of the tetrameric population is double that of the dimeric one, and both are within the experimental error of the MALS measurement.” (page 6, line 123).

4. Have the authors run the peaks in Fig 4b using BN-PAGE? It is important to note that the authors did not purify the TTYH proteins in this experiment, so the different peaks seen in the FSEC might be some complexes formed by the TTYH and other unknown proteins. Thus, it is important to, at the bare-minimum, analyze the peaks using BN-PAGE and Western. From the FSEC profile, the peak assumed to be the dimer of TTYH3 is about 1 ml earlier than the peak assumed to be the dimer of TTYH1. However, from Fig 4a, the size difference between dimer-TTYH1 and dimer-TTYH3 is less than 30kD. The elution volume difference seems to be too large. In addition, if the authors analyze the peaks assumed to be TTYH tetramers using BN-PAGE and Western and only see a tetramer band, it will rule out the possibility that “the dimeric population may arise from the reagents used for the BN-PAGE analysis”.

Response: We have not performed BN-PAGE analysis of the proteins analyzed by FSEC presented in Fig. 4b (revised Fig. 3b). Notably, BN-PAGE analysis of the peaks cannot rule out complex formation with additional proteins, as only mTTYH is identified using specific antibodies in this case. To alleviate the reviewer’s concerns, we purified the mTTYH1-GFP complex and performed SEC-MALS analysis (revised Fig. 3c, appended on page 5). Importantly, the elution profile of the purified protein is essentially the same as that obtained from whole-cell lysates. Furthermore, the MALS analysis supports the notion that the two peaks correspond to tetramers and dimers.

5. Because it had not been convincingly established that the peaks in Fig 4b correspond to dimeric and tetrameric TTYHs, Figure 5 and Figure 6 become less meaningful as well. For the HDX-MS experiment, the authors should include LC-MS quality control and a list of all the peptides/proteins identified from the mass spec experiment from the MASCOT search including peptide count, peptide score, p-values, etc. as supplementary data. How many times have the authors done the HDX-MS experiment at 4 degrees? Only one result is shown in the paper. I believe at least 1 more replication needs to be included as supplementary data for the result to be convincing.

Response: We hope that the new SEC-MALS data convincingly establish that the peaks in Fig. 4b (revised Fig. 3b, appended on page 5) correspond to tetramers and dimers.

Per the reporting of HDX-MS-related data, these have been deposited to the ProteomeXchange Consortium via the PRIDE partner repository with the dataset identifier PXD031833. These data will become publicly available upon publication. In the meantime, the reviewers may access the deposition through the following link: <http://www.ebi.ac.uk/pride> (Username: reviewer_pxd031833@ebi.ac.uk; Password: ofkrZcPf) Finally, the number of replications is indicated in the Materials and Methods section.

6. From the HDX-MS result, the authors identified a disulfide bond (C303-C370) that could contribute to tetramerization. However, by treating the protein with 2-ME, all disulfide bonds in all proteins are reduced. Thus, one cannot conclude that the assumed tetramer-to-dimer shift is due to the break of “intra-subunit disulfides” as the subtitle suggests. A more appropriate experiment is to mutate either C303 or C370 to alanine to only break that one disulfide bond and test whether there is a shift towards the assumed dimer peak.

Response: In the revised manuscript we provide a new FSEC analysis of the mTTYH3 mutants C299A and C366A. This analysis reveals a major decrease in the tetramer/dimer ratio, reminiscent of the effect of 2-ME. These results appear in Fig. 5d-f (appended on page 12).

“To further pinpoint the disulfide bridge contributing to tetramer stabilization, we proceeded with analyses of mTTYH3-GFP ECD cysteine mutants' impact on tetramerization (Fig. 5d). Akin to the effect of 2-ME, the FSEC elution profile of C299A and C366A (Fig. 5e) resulted in a marked shift towards dimers (Fig. 5f).” (page 9, line 200).

“...Indeed, targeting one of the ECD intra-subunit disulfide bridges by site-directed mutagenesis, also resulted in a low tetramer/dimer ratio (Fig. 5e, f).” (page 13, line 288).

Unfortunately, the corresponding mTTYH1 mutants failed to express in several attempts, precluding a similar analysis for this paralog.

Revised Fig. 5

7. Could the authors explain in more detail why they express the engineered mTTYH3 ECDs as a secreted protein using insect cells and why they cross-link the secreted ECDs? I believe that the point of this study is to prove that in native conditions, the TTYHs form tetramers on cell membrane. The experiment in Figure 7 is neither native nor on the cell membrane. I don't believe it is necessary and particularly meaningful.

Response: We chose this experimental paradigm as the ECD is a glycoprotein that is situated outside the cell. Thus, to allow proper intracellular processing of its post-translational modifications, the use of the secreted expression is needed for trafficking through the different organelles involved. Since our HDX-MS results support the role of the ECD in tetramerization, these experiments were performed to provide an orthogonal experimental proof of this notion. This is now better explained in the revised manuscript:

“Due to the presence of disulfide bridges and glycosyl moieties^{14,15,33}, we expressed this FLAG-tagged construct as a secreted protein in insect cells, to allow intracellular trafficking for proper post-translational modifications processing.” (Page 10, line 211).

Reviewer #4:

The manuscript authored by Melvin et al tries to demonstrate the tetrameric organization of Tweety homolog (TTYH) family proteins in native cellular membrane where they are potentially display and form Cl⁻ conducting channels in the membranes. The authors employ independent methodologies of crosslinking, superresolution microscopy, FSEC and HDX-MS to demonstrate the oligomerization in the case of TTYH family proteins. They generally conclude that detergent extraction destabilizes the tetramer and leads to dimeric arrangement as visualized in recent cryoEM structures. I have the following major concerns about this study.

Response: We thank Reviewer #4 for the helpful comments.

In the revised manuscript, we made efforts to include all of the Reviewer's suggestions. Please see below our detailed response.

1. While the authors seem to have done a thorough analysis of TTYH clustering using dSTORM analysis and do obtain a distribution that indicates tetramers of TTYH in the membrane, it was not clear how the bleach step assignment was done in Fig. 2e. The steps in this are not very clear.

Response: As indicated in the Materials and Methods section, bleaching steps were detected manually. This is the common practice in this field. Notably, the tetrameric KCNH1 was used as a positive control (Fig. 2g, h, i). In further support of the tetrameric stoichiometry of mTTYH, we now provide SEC-MALS analysis of purified mTTYH1, revealing a protein mass of 223 kDa, within the experimental error of the MALS measurement (revised Fig. 3c, appended on page 5).

“Next, to validate the oligomeric state of TTYH in each FSEC elution peak, we proceeded with purification of intact mTTYH1-GFP for SEC-MALS analysis (Fig. 3c). Indeed, the peaks eluting at the volumes assumed to represent tetramers and dimers revealed a protein mass of 223 ± 0.4 and 109 ± 1.4 kDa for the tetrameric and dimeric populations, respectively. As expected, the mass of the tetrameric population is double that of the dimeric one, and both are within the experimental error of the MALS measurement.” (page 6, line 123).

2. Given the expertise of the authors in HDX-MS based on the deuterium exchange pattern, can the authors model a tetrameric form of the TTYH1 and TTYH3 in a way that can satisfy the contact analyses.

Response: We thank the reviewer for raising this important point. Our data clearly support the tetrameric stoichiometry of mTTYH proteins in their native environment. However, unfortunately, we cannot derive the high-resolution spatial information needed to generate a tetrameric model, as suggested. Specifically, the HDX-MS data provide residue-level backbone exchange information without any spatial data, including contacts (these can be obtained by other methods, such as cross-linking MS, which were not utilized here). Moreover, as our HDX-MS analysis indicates that detergent solubilization may adversely affect the conformation of mTTYH subunits, using the dimeric structures as a starting point for further modeling of the tetrameric organization may result in highly inaccurate models. Therefore, while we agree that obtaining

the tetrameric structure is highly important for the field, this problem should be addressed experimentally in the future.

3. Related to the above, the notion of disulfides mediating tetramer formation is hTTYH1 in figure 6. One would have expected the authors to use a detergent system that allows formation of a tetramer in case of TTYH1, like LMNG, to perform the 2-mercaptoethanol exposure. Using DDM/CHS does not help tetramer formation (dashed line) in this case and once cannot conclude clear tetramer/dimer formation. This experiments clearly needs to be redone.

Response: Thank you for this insightful comment. As suggested, we repeated the FSEC analysis of the effect of 2-mercaptoethanol (2-ME) exposure following solubilization with LMNG. These results are presented in the newly added Supplementary Fig. 6. Importantly, both mTTYH1 and mTTYH3 show a clear decrease in the tetramer/dimer ratio, in line with the experiment performed using DDM (please note, DDM/CHS was erroneously indicated – this is corrected in the revised manuscript).

4. Related to the above, it would have been interesting to see how the profile shifts with the inclusion of different lipids including CHS, POPC, native lipid extracts to see effect on oligomerization. Contrary to the findings in this manuscript, the cryoEM structures reported by the Dutzler group determines the dimeric structures of the TTYH molecule within nanodiscs and GDN. One would presume that the TTYH oligomers in nanodiscs would be in the native tetrameric form if this is accurate.

Response: The process of reconstitution into nanodiscs requires the prior detergent-mediated solubilization of the membrane protein preparation. Therefore, the inclusion of lipids following the introduction of detergents does not mimic the native membrane conditions. Indeed, we did not observe an equilibrium between the dimeric and tetrameric populations, suggesting that the effect of detergents is irreversible.

5. Can the authors demonstrate the presence of any Cl⁻ conductance in HEK cells expressing the TTYH orthologues studied in this manuscript.

Response: As requested, we tested the presence of any Cl⁻ conductance in HEK 293 cells expressing our mTTYH3-GFP construct (identical to that used for FSEC). As can be seen in the representative current traces appended below, no significant current could be detected compared to a GFP control (n=5).

Of note, as similar mTTYH electrophysiological analyses were recently published by Li et al., Nat. Comm., 2021, we decided to refrain from redundancy and present these results only in the Response to Reviewers section.

6. Given that LMNG and DMNG extract tetrameric forms of the molecule, can the authors demonstrate negative stain EM or reference free 2D classes in cryoEM to validate their claim. This would provide a very clear proof of their findings.

Response: While the structural elucidation of the tetrameric complex is clearly of great interest to the field, such an endeavor is beyond the scope of the current manuscript. Moreover, the required resources to perform such experiments are also not readily available to us at this point.

Minor concerns.

Page 3 line 1. Represent scientific names accurately. *Drosophila melanogaster* and use capital D for *drosophila*.

Response: Corrected, thank you.

“The Tweety protein (TTY) was originally identified in the Drosophila melanogaster flightless locus^{1,2}. Subsequent studies identified three conserved TTY homologs (TTYH1-3) in vertebrates³⁻⁵, exhibiting a differential expression pattern.” ... (page 3, line 33).

REVIEWERS' COMMENTS:

Reviewer #1 (Remarks to the Author):

Authors addressed the comments adequately in the revised manuscript.

Reviewer #2 (Remarks to the Author):

All my questions and issues have been properly addressed. I support the publication of the manuscript in its present form.

Reviewer #3 (Remarks to the Author):

The authors have adequately addressed my questions and concerns by removing the STORM experiment results, performing new experiments, and revising their manuscripts.

One last thing I ask for the authors, as I mentioned in my comment last time, is to please upload the unprocessed Western films for Figure 1, Figure 3a, Figure 6b, and c as source data. Currently I can only see the original film for Supplementary Figure 1 in the source data.

Once the authors have uploaded all the asked images, I support the publication of this manuscript.

Reviewer #4 (Remarks to the Author):

All my comments for revision have been suitably addressed by the authors and can be published by the journal.